# Risk-Sensitive and Robust Decision-Making: a CVaR Optimization Approach

**Yinlam Chow**
Stanford University
ychow@stanford.edu

**Aviv Tamar**
UC Berkeley
avivt@berkeley.edu

**Shie Mannor**
Technion
shie@ee.technion.ac.il

**Marco Pavone**
Stanford University
pavone@stanford.edu

## Abstract

In this paper we address the problem of decision making within a Markov decision process (MDP) framework where risk and modeling errors are taken into account. Our approach is to minimize a risk-sensitive conditional-value-at-risk (CVaR) objective, as opposed to a standard risk-neutral expectation. We refer to such problem as CVaR MDP. Our first contribution is to show that a CVaR objective, besides capturing risk sensitivity, has an alternative interpretation as expected cost under worst-case modeling errors, for a given error budget. This result, which is of independent interest, motivates CVaR MDPs as a unifying framework for risk-sensitive and robust decision making. Our second contribution is to present an approximate value-iteration algorithm for CVaR MDPs and analyze its convergence rate. To our knowledge, this is the first solution algorithm for CVaR MDPs that enjoys error guarantees. Finally, we present results from numerical experiments that corroborate our theoretical findings and show the practicality of our approach.

## 1 Introduction

Decision making within the Markov decision process (MDP) framework typically involves the minimization of a risk-neutral performance objective, namely the *expected* total discounted cost [3]. This approach, while very popular, natural, and attractive from a computational standpoint, neither takes into account the *variability* of the cost (i.e., fluctuations around the mean), nor its *sensitivity* to modeling errors, which may significantly affect overall performance [12]. Risk-sensitive MDPs [9] address the first aspect by replacing the risk-neutral expectation with a *risk-measure* of the total discounted cost, such as variance, Value-at-Risk (VaR), or Conditional-VaR (CVaR). Robust MDPs [15], on the other hand, address the second aspect by defining a set of plausible MDP parameters, and optimize decision with respect to the expected cost under worst-case parameters.

In this work we consider risk-sensitive MDPs with a CVaR objective, referred to as CVaR MDPs. CVaR [1, 20] is a risk-measure that is rapidly gaining popularity in various engineering applications, e.g., finance, due to its favorable computational properties [1] and superior ability to safeguard a decision maker from the "outcomes that hurt the most" [22]. In this paper, by *relating risk to robustness*, we derive a novel result that further motivates the usage of a CVaR objective in a decision-making context. Specifically, we show that the CVaR of a discounted cost in an MDP is *equivalent* to the expected value of the same discounted cost in presence of worst-case perturbations of the MDP parameters (specifically, transition probabilities), provided that such perturbations are within a certain error budget. This result suggests CVaR MDP as a method for decision making under *both* cost variability *and* model uncertainty, motivating it as *unified framework* for planning under uncertainty.

*Literature review*: Risk-sensitive MDPs have been studied for over four decades, with earlier efforts focusing on exponential utility [9], mean-variance [24], and percentile risk criteria [7] . Recently, for the reasons explained above, several authors have investigated CVaR MDPs [20]. Specifically,

in [4], the authors propose a dynamic programming algorithm for finite-horizon risk-constrained MDPs where risk is measured according to CVaR. The algorithm is proven to asymptotically converge to an optimal risk-constrained policy. However, the algorithm involves computing integrals over continuous variables (Algorithm 1 in [4]) and, in general, its implementation appears particularly difficult. In [2], the authors investigate the structure of CVaR optimal policies and show that a Markov policy is optimal on an augmented state space, where the additional (continuous) state variable is represented by the running cost. In [8], the authors leverage such result to design an algorithm for CVaR MDPs that relies on discretizing occupation measures in the augmented-state MDP. This approach, however, involves solving a non-convex program via a sequence of linear-programming approximations, which can only shown to converge asymptotically. A different approach is taken by [5], [19] and [25], which consider a finite dimensional parameterization of control policies, and show that a CVaR MDP can be optimized to a *local* optimum using stochastic gradient descent (policy gradient). A recent result by Pflug and Pichler [17] showed that CVaR MDPs admit a dynamic programming formulation by using a state-augmentation procedure different from the one in [2]. The augmented state is also continuous, making the design of a solution algorithm challenging.

*Contributions*: The contribution of this paper is twofold. First, as discussed above, we provide a novel interpretation for CVaR MDPs in terms of robustness to modeling errors. This result is of independent interest and further motivates the usage of CVaR MDPs for decision making under uncertainty. Second, we provide a new optimization algorithm for CVaR MDPs, which leverages the state augmentation procedure introduced by Pflug and Pichler [17]. We overcome the aforementioned computational challenges (due to the continuous augmented state) by designing an algorithm that merges approximate value iteration [3] with linear interpolation. Remarkably, we are able to provide explicit error bounds and convergence rates based on contraction-style arguments. In contrast to the algorithms in [4, 8, 5, 25], given the explicit MDP model our approach leads to finite-time error guarantees, with respect to the *globally* optimal policy. In addition, our algorithm is significantly simpler than previous methods, and calculates the optimal policy *for all* CVaR confidence intervals and initial states simultaneously. The practicality of our approach is demonstrated in numerical experiments involving planning a path on a grid with thousand of states. To the best of our knowledge, this is the first algorithm to approximate globally-optimal policies for non-trivial CVaR MDPs whose error depends on the resolution of interpolation.

*Organization*: This paper is structured as follows. In Section 2 we provide background on CVaR and MDPs, we state the problem we wish to solve (i.e., CVaR MDPs), and motivate the CVaR MDP formulation by establishing a novel relation between CVaR and model perturbations. Section 3 provides the basis for our solution algorithm, based on a Bellman-style equation for the CVaR. Then, in Section 4 we present our algorithm and correctness analysis. In Section 5 we evaluate our approach via numerical experiments. Finally, in Section 6, we draw some conclusions and discuss directions for future work.

## 2 Preliminaries, Problem Formulation, and Motivation

### 2.1 Conditional Value-at-Risk

Let $Z$ be a bounded-mean random variable, i.e., $\mathbb{E}[|Z|] < \infty$, on a probability space $(\Omega, \mathcal{F}, \mathbb{P})$, with cumulative distribution function $F(z) = \mathbb{P}(Z \leq z)$. In this paper we interpret $Z$ as a cost. The *value-at-risk* (VaR) at confidence level $\alpha \in (0, 1)$ is the $1 - \alpha$ quantile of $Z$, i.e., $\text{VaR}_\alpha(Z) = \min\{z \mid F(z) \geq 1 - \alpha\}$. The *conditional value-at-risk* (CVaR) at confidence level $\alpha \in (0, 1)$ is defined as [20]:

$$\text{CVaR}_\alpha(Z) = \min_{w \in \mathbb{R}} \left\{ w + \frac{1}{\alpha} \mathbb{E}\big[(Z - w)^+\big] \right\}, \tag{1}$$

where $(x)^+ = \max(x, 0)$ represents the positive part of $x$. If there is no probability atom at $\text{VaR}_\alpha(Z)$, it is well known from Theorem 6.2 in [23] that $\text{CVaR}_\alpha(Z) = \mathbb{E}\big[Z \mid Z \geq \text{VaR}_\alpha(Z)\big]$. Therefore, $\text{CVaR}_\alpha(Z)$ may be interpreted as the expected value of $Z$, conditioned on the $\alpha$-portion of the tail distribution. It is well known that $\text{CVaR}_\alpha(Z)$ is decreasing in $\alpha$, $\text{CVaR}_1(Z)$ equals to $\mathbb{E}(Z)$, and $\text{CVaR}_\alpha(Z)$ tends to $\max(Z)$ as $\alpha \downarrow 0$. During the last decade, the CVaR risk-measure has gained popularity in financial applications, among others. It is especially useful for controlling rare, but potentially disastrous events, which occur above the $1 - \alpha$ quantile, and are neglected by the VaR [22]. Furthermore, CVaR enjoys desirable axiomatic properties, such as coherence [1]. We refer to [26] for further motivation about CVaR and a comparison with other risk measures such as VaR.

A useful property of CVaR, which we exploit in this paper, is its alternative dual representation [1]:

$$\text{CVaR}_\alpha(Z) = \max_{\xi \in \mathcal{U}_{\text{CVaR}}(\alpha, \mathbb{P})} \mathbb{E}_\xi[Z], \tag{2}$$

where $\mathbb{E}_\xi[Z]$ denotes the $\xi$-weighted expectation of $Z$, and the *risk envelope* $\mathcal{U}_{\text{CVaR}}$ is given by
$\mathcal{U}_{\text{CVaR}}(\alpha, \mathbb{P}) = \left\{ \xi : \xi(\omega) \in \left[0, \frac{1}{\alpha}\right], \int_{\omega \in \Omega} \xi(\omega)\mathbb{P}(\omega)d\omega = 1 \right\}$. Thus, the CVaR of a random variable $Z$ may be interpreted as the worst-case expectation of $Z$, under a perturbed distribution $\xi\mathbb{P}$.

In this paper, we are interested in the CVaR of the total discounted cost in a sequential decision-making setting, as discussed next.

## 2.2 Markov Decision Processes

An MDP is a tuple $\mathcal{M} = (\mathcal{X}, \mathcal{A}, C, P, x_0, \gamma)$, where $\mathcal{X}$ and $\mathcal{A}$ are finite state and action spaces; $C(x, a) \in [-C_{\max}, C_{\max}]$ is a bounded deterministic cost; $P(\cdot|x, a)$ is the transition probability distribution; $\gamma \in [0, 1)$ is the discounting factor, and $x_0$ is the initial state. (Our results easily generalize to random initial states and random costs.)

Let the space of admissible histories up to time $t$ be $H_t = H_{t-1} \times \mathcal{A} \times \mathcal{X}$, for $t \geq 1$, and $H_0 = \mathcal{X}$. A generic element $h_t \in H_t$ is of the form $h_t = (x_0, a_0, \ldots, x_{t-1}, a_{t-1}, x_t)$. Let $\Pi_{H,t}$ be the set of all history-dependent policies with the property that at each time $t$ the randomized control action is a function of $h_t$. In other words, $\Pi_{H,t} := \big\{ \mu_0 : H_0 \to \mathbb{P}(\mathcal{A}), \mu_1 : H_1 \to \mathbb{P}(\mathcal{A}), \ldots, \mu_t : H_t \to \mathbb{P}(\mathcal{A}) \big\} | \mu_j(h_j) \in \mathbb{P}(\mathcal{A})$ for all $h_j \in H_j, 1 \leq j \leq t\big\}$. We also let $\Pi_H = \lim_{t \to \infty} \Pi_{H,t}$ be the set of all history dependent policies.

## 2.3 Problem Formulation

Let $C(x_t, a_t)$ denote the stage-wise costs observed along a state/control trajectory in the MDP model, and let $\mathcal{C}_{0,T} = \sum_{t=0}^{T} \gamma^t C(x_t, a_t)$ denote the total discounted cost up to time $T$. The risk-sensitive discounted-cost problem we wish to address is as follows:

$$\min_{\mu \in \Pi_H} \quad \text{CVaR}_\alpha \left( \lim_{T \to \infty} \mathcal{C}_{0,T} \,\Big|\, x_0, \mu \right), \tag{3}$$

where $\mu = \{\mu_0, \mu_1, \ldots\}$ is the policy sequence with actions $a_t = \mu_t(h_t)$ for $t \in \{0, 1, \ldots\}$. We refer to problem (3) as CVaR MDP (One may also consider a related formulation combining mean and CVaR, the details of which are presented in the supplementary material).

The problem formulation in (3) directly addresses the aspect of risk sensitivity, as demonstrated by the numerous applications of CVaR optimization in finance (see, e.g., [21, 11, 6]) and the recent approaches for CVaR optimization in MDPs [4, 8, 5, 25]. In the following, we show a new result providing additional motivation for CVaR MDPs, from the point of view of *robustness to modeling errors*.

## 2.4 Motivation - Robustness to Modeling Errors

We show a new result relating the CVaR objective in (3) to the *expected* discounted-cost in presence of worst-case perturbations of the MDP parameters, where the perturbations are budgeted according to the "number of things that can go wrong". Thus, by minimizing CVaR, the decision maker also guarantees *robustness* of the policy.

Consider a trajectory $(x_0, a_0, \ldots, x_T)$ in a finite-horizon MDP problem with transitions $P_t(x_t|x_{t-1}, a_{t-1})$. We explicitly denote the time index of the transition matrices for reasons that will become clear shortly. The total probability of the trajectory is $P(x_0, a_0, \ldots, x_T) = P_0(x_0)P_1(x_1|x_0, a_0) \cdots P_T(x_T|x_{T-1}, a_{T-1})$, and we let $\mathcal{C}_{0,T}(x_0, a_0, \ldots, x_T)$ denote its discounted cost, as defined above.

We consider an adversarial setting, where an adversary is allowed to change the transition probabilities at each stage, under some budget constraints. We will show that, for a specific budget and perturbation structure, the expected cost under the worst-case perturbation is equivalent to the CVaR of the cost. Thus, we shall establish that, in this perspective, being risk sensitive is *equivalent* to being robust against model perturbations.

For each stage $1 \leq t \leq T$, consider a perturbed transition matrix $\hat{P}_t = P_t \circ \delta_t$, where $\delta_t \in \mathbb{R}^{\mathcal{X} \times \mathcal{A} \times \mathcal{X}}$ is a *multiplicative probability perturbation* and $\circ$ is the Hadamard product, under the condition that $\hat{P}_t$ is a stochastic matrix. Let $\Delta_t$ denote the set of perturbation matrices that satisfy this condition, and let $\Delta = \Delta_1 \times \cdots \times \Delta_T$ the set of all possible perturbations to the trajectory distribution.

We now impose a budget constraint on the perturbations as follows. For some budget $\eta \geq 1$, we consider the constraint

$$\delta_1(x_1|x_0, a_0)\delta_2(x_2|x_1, a_1) \cdots \delta_T(x_T|x_{T-1}, a_{T-1}) \leq \eta, \quad \forall x_0, \ldots, x_T \in \mathcal{X}, \forall a_0, \ldots, a_{T-1} \in \mathcal{A}. \tag{4}$$

Essentially, the product in Eq. (4) states that with small budget *the worst cannot happen at each time*. Instead, the perturbation budget has to be split (multiplicatively) along the trajectory. We note that Eq. (4) is in fact a constraint on the perturbation matrices, and we denote by $\Delta_\eta \subset \Delta$ the set of perturbations that satisfy this constraint with budget $\eta$. The following result shows an equivalence between the CVaR and the worst-case expected loss.

**Proposition 1 (Interpretation of CVaR as a Robustness Measure)** *It holds*

$$CVaR_{\frac{1}{\eta}} \left( \mathcal{C}_{0,T}(x_0, a_0, \ldots, x_T) \right) = \sup_{(\delta_1, \ldots, \delta_T) \in \Delta_\eta} \mathbb{E}_{\hat{P}} \left[ \mathcal{C}_{0,T}(x_0, a_0, \ldots, x_T) \right], \tag{5}$$

*where $\mathbb{E}_{\hat{P}}[\cdot]$ denotes expectation with respect to a Markov chain with transitions $\hat{P}_t$.*

The proof of Proposition 1 is in the supplementary material. It is instructive to compare Proposition 1 with the dual representation of CVaR in (2) where both results convert the CVaR risk into a robustness measure. Note, in particular, that the perturbation budget in Proposition 1 has a *temporal* structure, which constrains the adversary from choosing the worst perturbation at each time step.

**Remark 1** *An equivalence between robustness and risk-sensitivity was previously suggested by Osogami [16]. In that study, the* iterated *(dynamic) coherent risk was shown to be equivalent to a robust MDP [10] with a rectangular uncertainty set. The iterated risk (and, correspondingly, the rectangular uncertainty set) is very conservative [27], in the sense that* the worst can happen at each time step. *In contrast, the perturbations considered here are much less conservative. In general, solving robust MDPs without the rectangularity assumption is NP-hard. Nevertheless, Mannor et. al. [13] showed that, for cases where the number of perturbations to the parameters along a trajectory is upper bounded (budget-constrained perturbation), the corresponding robust MDP problem is tractable. Analogous to the constraint set (1) in [13], the perturbation set in Proposition 1 limits the total number of log-perturbations along a trajectory. Accordingly, we shall later see that optimizing problem* (3) *with perturbation structure* (4) *is indeed also tractable.*

Next section provides the fundamental theoretical ideas behind our approach to the solution of (3).

## 3 Bellman Equation for CVaR

In this section, by leveraging a recent result from [17], we present a dynamic programming (DP) formulation for the CVaR MDP problem in (3). As we shall see, the value function in this formulation depends on both the state and the CVaR confidence level $\alpha$. We then establish important properties of such DP formulation, which will later enable us to derive an efficient DP-based approximate solution algorithm and provide correctness guarantees on the approximation error. All proofs are presented in the supplementary material.

Our starting point is a recursive decomposition of CVaR, whose proof is detailed in Theorem 10 of [17].

**Theorem 2 (CVaR Decomposition, Theorem 21 in [17])** *For any $t \geq 0$, denote by $Z = (Z_{t+1}, Z_{t+2}, \ldots)$ the cost sequence from time $t + 1$ onwards. The conditional CVaR under policy $\mu$, i.e., $CVaR_\alpha(Z \mid h_t, \mu)$, obeys the following decomposition:*

$$CVaR_\alpha(Z \mid h_t, \mu) = \max_{\xi \in \mathcal{U}_{CVaR}(\alpha, P(\cdot|x_t, a_t))} \mathbb{E}[\xi(x_{t+1}) \cdot CVaR_{\alpha\xi(x_{t+1})}(Z \mid h_{t+1}, \mu) \mid h_t, \mu],$$

*where $a_t$ is the action induced by policy $\mu_t(h_t)$, and the expectation is with respect to $x_{t+1}$.*

Theorem 2 concerns a fixed policy $\mu$; we now extend it to a general DP formulation. Note that in the recursive decomposition in Theorem 2 the right-hand side involves CVaR terms with different confidence levels than that in the left-hand side. Accordingly, we augment the state space $\mathcal{X}$ with an additional continuous state $\mathcal{Y} = (0, 1]$, which corresponds to the confidence level. For any $x \in \mathcal{X}$ and $y \in \mathcal{Y}$, the *value-function* $V(x, y)$ for the augmented state $(x, y)$ is defined as:

$$V(x, y) = \min_{\mu \in \Pi_H} CVaR_y \left( \lim_{T \to \infty} \mathcal{C}_{0,T} \mid x_0 = x, \mu \right).$$

Similar to standard DP, it is convenient to work with operators defined on the space of value functions [3]. In our case, Theorem 2 leads to the following definition of CVaR Bellman operator $\mathbf{T} : \mathcal{X} \times \mathcal{Y} \to \mathcal{X} \times \mathcal{Y}$:

$$\mathbf{T}[V](x,y) = \min_{a \in \mathcal{A}} \left[ C(x,a) + \gamma \max_{\xi \in \mathcal{U}_{\text{CVaR}}(y, P(\cdot|x,a))} \sum_{x' \in \mathcal{X}} \xi(x') V\left(x', y\xi(x')\right) P(x'|x,a) \right]. \quad (6)$$

We now establish several useful properties for the Bellman operator $\mathbf{T}[V]$.

**Lemma 3 (Properties of CVaR Bellman Operator)** *The Bellman operator $\mathbf{T}[V]$ has the following properties:*

1. *(Contraction.) $\|\mathbf{T}[V_1] - \mathbf{T}[V_2]\|_\infty \le \gamma \|V_1 - V_2\|_\infty$, where $\|f\|_\infty = \sup_{x \in \mathcal{X}, y \in \mathcal{Y}} |f(x,y)|$.*

2. *(Concavity preserving in $y$.) For any $x \in \mathcal{X}$, suppose $yV(x,y)$ is concave in $y \in \mathcal{Y}$. Then the maximization problem in (6) is concave. Furthermore, $y\mathbf{T}[V](x,y)$ is concave in $y$.*

The first property in Lemma 3 is similar to standard DP [3], and is instrumental to the design of a converging value-iteration approach. The second property is nonstandard and specific to our approach. It will be used to show that the computation of value-iteration updates involves concave, and therefore *tractable* optimization problems. Furthermore, it will be used to show that a linear-interpolation of $V(x,y)$ in the augmented state $y$ has a bounded error.

Equipped with the results in Theorem 2 and Lemma 3, we can now show that the fixed point solution of $\mathbf{T}[V](x,y) = V(x,y)$ is unique, and equals to the solution of the CVaR MDP problem (3) with $x_0 = x$ and $\alpha = y$.

**Theorem 4 (Optimality Condition)** *For any $x \in \mathcal{X}$ and $y \in (0,1]$, the solution to $\mathbf{T}[V](x,y) = V(x,y)$ is unique, and equals to $V^*(x,y) = \min_{\mu \in \Pi_H} \text{CVaR}_y (\lim_{T \to \infty} \mathcal{C}_{0,T} \mid x_0 = x, \mu)$.*

Next, we show that the optimal value of the CVaR MDP problem (3) can be attained by a stationary Markov policy, defined as a greedy policy with respect to the value function $V^*(x,y)$. Thus, while the original problem is defined over the intractable space of history-dependent policies, a stationary Markov policy (over the augmented state space) is optimal, and can be readily derived from $V^*(x,y)$. Furthermore, an optimal history-dependent policy can be readily obtained from an (augmented) optimal Markov policy according to the following theorem.

**Theorem 5 (Optimal Policies)** *Let $\pi_H^* = \{\mu_0, \mu_1, \ldots\} \in \Pi_H$ be a history-dependent policy recursively defined as:*

$$\mu_k(h_k) = u^*(x_k, y_k), \ \forall k \ge 0, \quad (7)$$

*with initial conditions $x_0$ and $y_0 = \alpha$, and state transitions*

$$x_k \sim P(\cdot \mid x_{k-1}, u^*(x_{k-1}, y_{k-1})), \quad y_k = y_{k-1}\xi^*_{x_{k-1}, y_{k-1}, u^*}(x_k), \forall k \ge 1, \quad (8)$$

*where the stationary Markovian policy $u^*(x,y)$ and risk factor $\xi^*_{x,y,u^*}(\cdot)$ are solution to the min-max optimization problem in the CVaR Bellman operator $\mathbf{T}[V^*](x,y)$. Then, $\pi_H^*$ is an optimal policy for problem (3) with initial state $x_0$ and CVaR confidence level $\alpha$.*

Theorems 4 and 5 suggest that a value-iteration DP method [3] can be used to solve the CVaR MDP problem (3). Let an initial value-function guess $V_0 : \mathcal{X} \times \mathcal{Y} \to \mathbb{R}$ be chosen arbitrarily. Value iteration proceeds recursively as follows:

$$V_{k+1}(x,y) = \mathbf{T}[V_k](x,y), \ \forall (x,y) \in \mathcal{X} \times \mathcal{Y}, \ k \in \{0, 1, \ldots, \}. \quad (9)$$

Specifically, by combining the contraction property in Lemma 3 and uniqueness result of fixed point solutions from Theorem 4, one concludes that $\lim_{k \to \infty} V_k(x,y) = V^*(x,y)$. By selecting $x = x_0$ and $y = \alpha$, one immediately obtains $V^*(x_0, \alpha) = \min_{\mu \in \Pi_H} \text{CVaR}_\alpha (\lim_{T \to \infty} \mathcal{C}_{0,T} \mid x_0, \mu)$. Furthermore, an optimal policy may be derived from $V^*(x,y)$ according to the policy construction procedure in Theorem 5.

Unfortunately, while value iteration is conceptually appealing, its direct implementation in our setting is generally impractical since, e.g., the state $y$ is continuous. In the following, we pursue an *approximation* to the value iteration algorithm (9), based on a linear interpolation scheme for $y$.

---
**Algorithm 1** `CVaR Value Iteration with Linear Interpolation`
---
1: **Given:**

- $N(x)$ interpolation points $\mathbf{Y}(x) = \{y_1, \ldots, y_{N(x)}\} \in [0,1]^{N(x)}$ for every $x \in \mathcal{X}$ with $y_i < y_{i+1}$, $y_1 = 0$ and $y_{N(x)} = 1$.

- Initial value function $V_0(x,y)$ that satisfies Assumption 1.

2: For $t = 1, 2, \ldots$

- For each $x \in \mathcal{X}$ and each $y_i \in \mathbf{Y}(x)$, update the value function estimate as follows:

$$V_t(x, y_i) = \mathbf{T}_{\mathcal{I}}[V_{t-1}](x, y_i),$$

3: Set the converged value iteration estimate as $\widehat{V}^*(x, y_i)$, for any $x \in \mathcal{X}$, and $y_i \in \mathbf{Y}(x)$.

---

## 4   Value Iteration with Linear Interpolation

In this section we present an approximate DP algorithm for solving CVaR MDPs, based on the theoretical results of Section 3. The value iteration algorithm in Eq. (9) presents two main implementation challenges. The first is due to the fact that the augmented state $y$ is continuous. We handle this challenge by using interpolation, and exploit the concavity of $yV(x,y)$ to bound the error introduced by this procedure. The second challenge stems from the the fact that applying $\mathbf{T}$ involves maximizing over $\xi$. Our strategy is to exploit the concavity of the maximization problem to guarantee that such optimization can indeed be performed effectively.

As discussed, our approach relies on the fact that the Bellman operator $\mathbf{T}$ preserves concavity as established in Lemma 3. Accordingly, we require the following assumption for the initial guess $V_0(x,y)$,

**Assumption 1** *The guess for the initial value function $V_0(x,y)$ satisfies the following properties: 1) $yV_0(x,y)$ is concave in $y \in \mathcal{Y}$ and 2) $V_0(x,y)$ is continuous in $y \in \mathcal{Y}$ for any $x \in \mathcal{X}$.*

Assumption 1 may easily be satisfied, for example, by choosing $V_0(x,y) = \text{CVaR}_y(Z \mid x_0 = x)$, where $Z$ is any arbitrary bounded random variable. As stated earlier, a key difficulty in applying value iteration (9) is that, for each state $x \in \mathcal{X}$, the Bellman operator has to be calculated for each $y \in \mathcal{Y}$, and $\mathcal{Y}$ is continuous. As an approximation, we propose to calculate the Bellman operator only for a finite set of values $y$, and interpolate the value function in between such interpolation points.

Formally, let $N(x)$ denote the number of interpolation points. For every $x \in \mathcal{X}$, denote by $\mathbf{Y}(x) = \{y_1, \ldots, y_{N(x)}\} \in [0,1]^{N(x)}$ the set of interpolation points. We denote by $\mathcal{I}_x[V](y)$ the linear interpolation of the function $yV(x,y)$ on these points, i.e.,

$$\mathcal{I}_x[V](y) = y_i V(x, y_i) + \frac{y_{i+1} V(x, y_{i+1}) - y_i V(x, y_i)}{y_{i+1} - y_i}(y - y_i),$$

where $y_i = \max\{y' \in \mathbf{Y}(x) : y' \leq y\}$ and $y_{i+1}$ is the closest interpolation point such that $y \in [y_i, y_{i+1}]$, i.e., $y_{i+1} = \min\{y' \in \mathbf{Y}(x) : y' \geq y\}$. The interpolation of $yV(x,y)$ instead of $V(x,y)$ is key to our approach. The motivation is twofold: first, it can be shown [20] that for a discrete random variable $Z$, $y\text{CVaR}_y(Z)$ is piecewise linear in $y$. Second, one can show that the Lipschitzness of $yV(x,y)$ is preserved during value iteration, and exploit this fact to bound the linear interpolation error.

We now define the *interpolated* Bellman operator $\mathbf{T}_{\mathcal{I}}$ as follows:

$$\mathbf{T}_{\mathcal{I}}[V](x,y) = \min_{a \in \mathcal{A}} \left[ C(x,a) + \gamma \max_{\xi \in \mathcal{U}_{\text{CVaR}}(y, P(\cdot|x,a))} \sum_{x' \in \mathcal{X}} \frac{\mathcal{I}_{x'}[V](y\xi(x'))}{y} P(x'|x,a) \right]. \qquad (10)$$

**Remark 2** *Notice that by L'Hospital's rule one has $\lim_{y \to 0} \mathcal{I}_x[V](y\xi(x))/y = V(x,0)\xi(x)$. This implies that at $y = 0$ the interpolated Bellman operator is equivalent to the original Bellman operator, i.e., $\mathbf{T}[V](x,0) = \min_{a \in \mathcal{A}} \{C(x,a) + \gamma \max_{x' \in \mathcal{X}: P(x'|x,a)>0} V(x',0)\} = \mathbf{T}_{\mathcal{I}}[V](x,0)$.*

Algorithm 1 presents CVaR value iteration with linear interpolation. The only difference between this algorithm and standard value iteration (9) is the linear interpolation procedure described above. In the following, we show that Algorithm 1 converges, and bound the error due to interpolation. We begin by showing that the useful properties established in Lemma 3 for the Bellman operator $\mathbf{T}$ extend to the interpolated Bellman operator $\mathbf{T}_{\mathcal{I}}$.

**Lemma 6 (Properties of Interpolated Bellman Operator)** $\mathbf{T}_{\mathcal{I}}[V]$ *has the same properties of* $\mathbf{T}[V]$ *as in Lemma 3, namely 1) contraction and 2) concavity preservation.*

Lemma 6 implies several important consequences for Algorithm 1. The first one is that the maximization problem in (10) is concave, and thus may be solved efficiently at each step. This guarantees that the algorithm is *tractable*. Second, the contraction property in Lemma 6 guarantees that Algorithm 1 converges, i.e., there exists a value function $\widehat{V}^* \in \mathbb{R}^{|\mathcal{X}| \times |\mathcal{Y}|}$ such that $\lim_{n \to \infty} \mathbf{T}_{\mathcal{I}}^n [V_0](x, y_i) = \widehat{V}^*(x, y_i)$. In addition, the convergence rate is geometric and equals to $\gamma$.

The following theorem provides an error bound between approximate value iteration and exact value iteration (3) in terms of the interpolation resolution.

**Theorem 7 (Convergence and Error Bound)** *Suppose the initial value function $V_0(x, y)$ satisfies Assumption 1 and let $\epsilon > 0$ be an error tolerance parameter. For any state $x \in \mathcal{X}$ and step $t \geq 0$, choose $y_2 > 0$ such that $V_t(x, y_2) - V_t(x, 0) \geq -\epsilon$ and update the interpolation points according to the logarithmic rule: $y_{i+1} = \theta y_i$, $\forall i \geq 2$, with uniform constant $\theta \geq 1$. Then, Algorithm 1 has the following error bound:*

$$0 \geq \widehat{V}^*(x_0, \alpha) - \min_{\mu \in \Pi_H} CVaR_\alpha \left( \lim_{T \to \infty} \mathcal{C}_{0,T} \mid x_0, \mu \right) \geq \frac{-\gamma}{1-\gamma} \mathbf{O}\left( (\theta - 1) + \epsilon \right),$$

*and the following finite time convergence error bound:*

$$\left| \mathbf{T}_{\mathcal{I}}^n [V_0](x_0, \alpha) - \min_{\mu \in \Pi_H} CVaR_\alpha \left( \lim_{T \to \infty} \mathcal{C}_{0,T} \mid x_0, \mu \right) \right| \leq \frac{\mathbf{O}\left( (\theta - 1) + \epsilon \right) + \mathbf{O}(\gamma^n)}{1 - \gamma}.$$

Theorem 7 shows that 1) the interpolation-based value function is a *conservative estimate* for the optimal solution to problem (3); 2) the interpolation procedure is *consistent*, i.e., when the number of interpolation points is arbitrarily large (specifically, $\epsilon \to 0$ and $y_{i+1}/y_i \to 1$), the approximation error tends to zero; and 3) the approximation error bound is $O((\theta - 1) + \epsilon)$, where $\log \theta$ is the *log-difference* of the interpolation points, i.e., $\log \theta = \log y_{i+1} - \log y_i$, $\forall i$.

For a pre-specified $\epsilon$, the condition $V_t(x, y_2) - V_t(x, 0) \geq -\epsilon$ may be satisfied by a simple *adaptive procedure* for selecting the interpolation points $\mathbf{Y}(x)$. At each iteration $t > 0$, after calculating $V_t(x, y_i)$ in Algorithm 1, at each state $x$ in which the condition does not hold, add a new interpolation point $y_2' = \frac{\epsilon y_2}{|V_t(x, y_2) - V_t(x, 0)|}$, and additional points between $y_2'$ and $y_2$ such that the condition $\log \theta \geq \log y_{i+1} - \log y_i$ is maintained. Since all the additional points belong to the segment $[0, y_2]$, the linearly interpolated $V_t(x, y_i)$ remains unchanged, and Algorithm 1 proceeds as is. For bounded costs and $\epsilon > 0$, the number of additional points required is bounded.

The full proof of Theorem 7 is detailed in the supplementary material; we highlight the main ideas and challenges involved. In the first part of the proof we bound, for all $t > 0$, the Lipschitz constant of $yV_t(x, y)$ in $y$. The key to this result is to show that the Bellman operator $\mathbf{T}$ preserves the Lipschitz property for $yV_t(x, y)$. Using the Lipschitz bound and the concavity of $yV_t(x, y)$, we then bound the error $\frac{\mathcal{I}_x[V_t](y)}{y} - V_t(x, y)$ for all $y$. The condition on $y_2$ is required for this bound to hold when $y \to 0$. Finally, we use this result to bound $\|\mathbf{T}_{\mathcal{I}}[V_t](x, y) - \mathbf{T}[V_t](x, y)\|_\infty$. The results of Theorem 7 follow from contraction arguments, similar to approximate dynamic programming [3].

## 5 Experiments

We validate Algorithm 1 on a rectangular grid world, where states represent grid points on a 2D terrain map. An agent (e.g., a robotic vehicle) starts in a safe region and its objective is to travel to a given destination. At each time step the agent can move to any of its four neighboring states. Due to sensing and control noise, however, with probability $\delta$ a move to a random neighboring state occurs. The stage-wise cost of each move until reaching the destination is 1, to account for fuel usage. In between the starting point and the destination there are a number of obstacles that the agent should avoid. Hitting an obstacle costs $M \gg 1$ and terminates the mission. The objective is to compute a *safe* (i.e., obstacle-free) path that is *fuel efficient*.

For our experiments, we choose a $64 \times 53$ grid-world (see Figure 1), for a total of 3,312 states. The destination is at position $(60, 2)$, and there are 80 obstacles plotted in yellow. By leveraging Theorem 7, we use 21 log-spaced interpolation points for Algorithm 1 in order to achieve a small value function error. We choose $\delta = 0.05$, and a discount factor $\gamma = 0.95$ for an effective horizon of 200 steps. Furthermore, we set the penalty cost equal to $M = 2/(1 - \gamma)$–such choice trades off high penalty for collisions and computational complexity (that increases as $M$ increases). For the

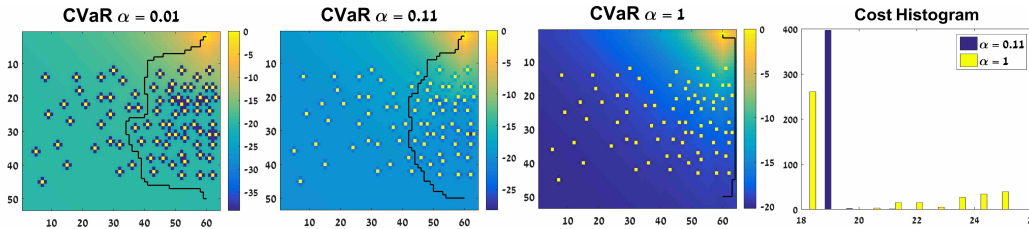

Figure 1: Grid-world simulation. Left three plots show the value functions and corresponding paths for different CVaR confidence levels. The rightmost plot shows a cost histogram (for 400 Monte Carlo trials) for a risk-neutral policy and a CVaR policy with confidence level $\alpha = 0.11$.

interpolation parameters discussed in Theorem 7, we set $\epsilon = 0.1$ and $\theta = 2.067$ (in order to have 21 logarithmically distributed grid points for the CVaR confidence parameter in $[0, 1]$).

In Figure 1 we plot the value function $V(x, y)$ for three different values of the CVaR confidence parameter $\alpha$, and the corresponding paths starting from the initial position $(60, 50)$. The first three figures in Figure 1 show how by decreasing the confidence parameter $\alpha$ the average travel distance (and hence fuel consumption) slightly increases but the collision probability decreases, as expected. We next discuss robustness to modeling errors. We conducted simulations in which with probability 0.5 each obstacle position is perturbed in a random direction to one of the neighboring grid cells. This emulates, for example, measurement errors in the terrain map. We then trained both the risk-averse ($\alpha = 0.11$) and risk-neutral ($\alpha = 1$) policies on the nominal (i.e., unperturbed) terrain map, and evaluated them on 400 perturbed scenarios (20 perturbed maps with 20 Monte Carlo evaluations each). While the risk-neutral policy finds a shorter route (with average cost equal to 18.137 on successful runs), it is vulnerable to perturbations and fails more often (with over 120 failed runs). In contrast, the risk-averse policy chooses slightly longer routes (with average cost equal to 18.878 on successful runs), but is much more robust to model perturbations (with only 5 failed runs).

For the computation of Algorithm 1 we represented the concave piecewise linear maximization problem in (10) as a linear program, and concatenated several problems to reduce repeated overhead stemming from the initialization of the CPLEX linear programming solver. This resulted in a computation time on the order of two hours. We believe there is ample room for improvement, for example by leveraging parallelization and sampling-based methods. Overall, we believe our proposed approach is currently the most practical method available for solving CVaR MDPs (as a comparison, the recently proposed method in [8] involves infinite dimensional optimization). The Matlab code used for the experiments is provided in the supplementary material.

## 6 Conclusion

In this paper we presented an algorithm for CVaR MDPs, based on approximate value-iteration on an augmented state space. We established convergence of our algorithm, and derived finite-time error bounds. These bounds are useful to stop the algorithm at a desired error threshold.

In addition, we uncovered an interesting relationship between the CVaR of the total cost and the worst-case expected cost under adversarial model perturbations. In this formulation, the perturbations are correlated in time, and lead to a robustness framework significantly less conservative than the popular robust-MDP framework, where the uncertainty is temporally independent.

Collectively, our work suggests CVaR MDPs as a unifying and practical framework for computing control policies that are robust with respect to both stochasticity and model perturbations. Future work should address extensions to large state-spaces. We conjecture that a sampling-based approximate DP approach [3] should be feasible since, as proven in this paper, the CVaR Bellman equation is contracting (as required by approximate DP methods).

**Acknowledgement**

The authors would like to thank Mohammad Ghavamzadeh for helpful comments on the technical details, and Daniel Vainsencher for practical optimization advice. Y.-L. Chow and M. Pavone are partially supported by the Croucher Foundation doctoral scholarship and the Office of Naval Research, Science of Autonomy Program, under Contract N00014-15-1-2673. Funding for Shie Mannor and Aviv Tamar were partially provided by the European Community's Seventh Framework Programme (FP7/2007-2013) under grant agreement 306638 (SUPREL).

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
