[Supplementary Material · cvarNIPS15.pdf]

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

# A Proofs of Theoretical Results

## A.1 Proof of Proposition 1

By definition, we have that

$$\mathbb{E}_{\hat{P}}\left[C(x_0, a_0, \ldots, x_T)\right]$$

$$= \sum_{(x_0, a_0, \ldots, x_T)} P_0(x_0)\delta_1(x_1|x_0, a_0)\cdots P_T(x_T|x_{T-1}, a_{T-1})\delta_T(x_T|x_{T-1}, a_{T-1})C(x_0, a_0, \ldots, x_T)$$

$$= \sum_{(x_0, a_0, \ldots, x_T)} P(x_0, a_0, \ldots, x_T)\delta_1(x_1|x_0, a_0)\delta_2(x_2|x_1, a_1)\cdots\delta_T(x_T|x_{T-1}, a_{T-1})C(x_0, a_0, \ldots, x_T)$$

$$\doteq \sum_{(x_0, a_0, \ldots, x_T)} P(x_0, a_0, \ldots, x_T)\delta(x_0, a_0, \ldots, x_T)C(x_0, a_0, \ldots, x_T).$$

Note that by definition of the set $\Delta$, for any $(\delta_1, \ldots, \delta_T) \in \Delta$ we have that $P(x_0, a_0, \ldots, x_T) > 0 \rightarrow \delta(x_0, a_0, \ldots, x_T) \geq 0$, and

$$\mathbb{E}\left[\delta(x_0, a_0, \ldots, x_T)\right] \doteq \sum_{(x_0, a_0, \ldots, x_T)} P(x_0, a_0, \ldots, x_T)\delta(x_0, a_0, \ldots, x_T) = 1.$$

Thus,

$$\sup_{(\delta_1, \ldots, \delta_T) \in \Delta_\eta} \mathbb{E}_{\hat{P}}\left[C(x_0, a_0, \ldots, x_T)\right]$$

$$= \sup_{\substack{0 \leq \delta(x_0, a_0, \ldots, x_T) \leq \eta, \\ \mathbb{E}[\delta(x_0, a_0, \ldots, x_T)] = 1}} \sum_{(x_0, a_0, \ldots, x_T)} P(x_0, a_0, \ldots, x_T)\delta(x_0, a_0, \ldots, x_T)C(x_0, a_0, \ldots, x_T) = \text{CVaR}_{\frac{1}{\eta}}\left(C(x_0, a_0, \ldots, x_T)\right),$$

where the last equality is by the representation theorem for CVaR [23].

## A.2 Proof of Lemma 3

The proof of monotonicity and constant shift properties follow directly from the definitions of the Bellman operator, by noting that $\xi(x')P(x'|x, a)$ is non-negative and $\sum_{x' \in \mathcal{X}} \xi(x')P(x'|x, a)] = 1$ for any $\xi \in \mathcal{U}_{\text{CVaR}}(y, P(\cdot|x, a))$. For the contraction property, denote $c = \|V_1 - V_2\|_\infty$. Since

$$V_2(x, y) - \|V_1 - V_2\|_\infty \leq V_1(x, y) \leq V_2(x, y) + \|V_1 - V_2\|_\infty, \; \forall x \in \mathcal{X}, \, y \in \mathcal{Y},$$

by monotonicity and constant shift property,

$$\mathbf{T}[V_2](x, y) - \gamma\|V_1 - V_2\|_\infty \leq \mathbf{T}[V_1](x, y) \leq \mathbf{T}[V_2](x, y) + \gamma\|V_1 - V_2\|_\infty \; \forall x \in \mathcal{X}, \, y \in \mathcal{Y}.$$

This further implies that

$$|\mathbf{T}[V_1](x, y) - \mathbf{T}[V_2](x, y)| \leq \gamma\|V_1 - V_2\|_\infty \; \forall x \in \mathcal{X}, \, y \in \mathcal{Y}$$

and the contraction property follows.

Now, we prove the concavity preserving property. Assume that $yV(x, y)$ is concave in $y$ for any $x \in \mathcal{X}$. Let $y_1, y_2 \in \mathcal{Y}$, and $\lambda \in [0, 1]$, and define $y_\lambda = (1 - \lambda)y_1 + \lambda y_2$. We have

$$(1 - \lambda)y_1\mathbf{T}[V](x, y_1) + \lambda y_2\mathbf{T}[V](x, y_2)$$

$$=(1 - \lambda)y_1 \min_{a_1 \in \mathcal{A}}\left[C(x, a_1) + \gamma \max_{\xi_1 \in \mathcal{U}_{\text{CVaR}}(y_1, P(\cdot|x, a_1))} \sum_{x' \in \mathcal{X}} \xi_1(x')V\left(x', y_1\xi_1(x')\right)P(x'|x, a_1)\right]$$

$$+ \lambda y_2 \min_{a_2 \in \mathcal{A}}\left[C(x, a_2) + \gamma \max_{\xi_2 \in \mathcal{U}_{\text{CVaR}}(y_2, P(\cdot|x, a_2))} \sum_{x' \in \mathcal{X}} \xi_2(x')V\left(x', y_2\xi_2(x')\right)P(x'|x, a_2)\right]$$

$$= \min_{a_1 \in \mathcal{A}}\left[(1 - \lambda)y_1 C(x, a_1) + \gamma \max_{\xi_1 \in \mathcal{U}_{\text{CVaR}}(y_1, P(\cdot|x, a_1))} \sum_{x' \in \mathcal{X}} \xi_1(x')V\left(x', y_1\xi_1(x')\right)P(x'|x, a_1)(1 - \lambda)y_1\right]$$

$$+ \min_{a_2 \in \mathcal{A}}\left[\lambda y_2 C(x, a_2) + \gamma \max_{\xi_2 \in \mathcal{U}_{\text{CVaR}}(y_2, P(\cdot|x, a_2))} \sum_{x' \in \mathcal{X}} \xi_2(x')V\left(x', y_2\xi_2(x')\right)P(x'|x, a_2)\lambda y_2\right]$$

$$\leq \min_{a \in \mathcal{A}} \left[ y_\lambda C(x,a) + \gamma \max_{\substack{\xi_1 \in \mathcal{U}_{\text{CVaR}}(y_1, P(\cdot|x,a)) \\ \xi_2 \in \mathcal{U}_{\text{CVaR}}(y_2, P(\cdot|x,a))}} \sum_{x' \in \mathcal{X}} P(x'|x,a) \left( (1-\lambda) y_1 \xi_1(x') V\left(x', y_1 \xi_1(x')\right) + \lambda y_2 \xi_2(x') V\left(x', y_2 \xi_2(x')\right) \right) \right]$$

$$\leq \min_{a \in \mathcal{A}} \left[ y_\lambda C(x,a) + \gamma \max_{\substack{\xi_1 \in \mathcal{U}_{\text{CVaR}}(y_1, P(\cdot|x,a)) \\ \xi_2 \in \mathcal{U}_{\text{CVaR}}(y_2, P(\cdot|x,a))}} \sum_{x' \in \mathcal{X}} P(x'|x,a) \left( (1-\lambda) y_1 \xi_1(x') + \lambda y_2 \xi_2(x') \right) V\left(x', ((1-\lambda) y_1 \xi_1(x') + \lambda y_2 \xi_2(x')) \right) \right]$$

where the first inequality is by concavity of the $\min$, and the second is by the concavity assumption. Now, define $\xi = \frac{(1-\lambda)y_1 \xi_1 + \lambda y_2 \xi_2}{y_\lambda}$. When $\xi_1 \in \mathcal{U}_{\text{CVaR}}(y_1, P(\cdot|x,a))$ and $\xi_2 \in \mathcal{U}_{\text{CVaR}}(y_2, P(\cdot|x,a))$, we have that $\xi \in \left[0, \frac{1}{y_\lambda}\right]$ and $\sum_{x' \in \mathcal{X}} \xi(x') \mathbb{P}(x'|x,a) = 1$. We thus have

$$(1-\lambda) y_1 \mathbf{T}[V](x, y_1) + \lambda y_2 \mathbf{T}[V](x, y_2)$$

$$\leq \min_{a \in \mathcal{A}} \left[ y_\lambda C(x,a) + \gamma \max_{\xi \in \mathcal{U}_{\text{CVaR}}(y_\lambda, P(\cdot|x,a))} \sum_{x' \in \mathcal{X}} P(x'|x,a) y_\lambda \xi(x') V\left(x', y_\lambda \xi(x')\right) \right]$$

$$= y_\lambda \min_{a \in \mathcal{A}} \left[ C(x,a) + \gamma \max_{\xi \in \mathcal{U}_{\text{CVaR}}(y_\lambda, P(\cdot|x,a))} \sum_{x' \in \mathcal{X}} P(x'|x,a) \xi(x') V\left(x', y_\lambda \xi(x')\right) \right] = y_\lambda \mathbf{T}[V](x, y_\lambda).$$

Finally, to show that the inner problem in (6) is a concave maximization, we need to show that

$$\Lambda_{x,y,a}(z) := \begin{cases} zV(x',z) P(x'|x,a)/y & \text{if } y \neq 0 \\ 0 & \text{otherwise} \end{cases}$$

is a concave function in $z \in \mathbb{R}$ for any given $x \in \mathcal{X}$, $y \in \mathcal{Y}$ and $a \in \mathcal{A}$. Suppose $zV(x,z)$ is a concave function in $z$. Immediately we can see that $\Lambda_{x,y,a}(z)$ is concave in $z$ when $y = 0$. Also notice that when $y \in \mathcal{Y} \setminus \{0\}$, since the transition probability $P(x'|x,a)$ is non-negative, we have the result that $\Lambda_{x,y,a}(z)$ is concave in $z$. This further implies

$$\sum_{x' \in \mathcal{X}} \frac{P(x'|x,a)}{y} \Lambda_{x,y,a}(y\xi(x')) = \sum_{x' \in \mathcal{X}} \xi(x') V(x', y\xi(x')) P(x'|x,a)$$

is concave in $\xi$. Furthermore by combining the result with the fact that the feasible set of $\xi$ is a polytope, we complete the proof of this claim.

## A.3 Proof of Theorem 4

The first part of the proof is to show that for any $(x,y) \in \mathcal{X} \times \mathcal{Y}$,
$$V_n(x,y) := \mathbf{T}^n[V_0](x,y) = \min_{\mu \in \Pi_M} \text{CVaR}_y \left( \mathcal{C}_{0,n} + \gamma^n V_0 \mid x_0 = x, \mu \right), \tag{11}$$

by induction, where the initial condition is $(x_0, y_0) = (x,y)$ and control action $a_t$ is induced by $\mu(x_t, y_t)$. For $n = 1$, we have that $V_1(x,y) = \mathbf{T}[V_0](x,y) = \min_{\mu \in \Pi_M} C(x_0, a_0) + \gamma \text{CVaR}_y \left( C(x_1, a_1) + V_0(x_1) \mid x_0 = x, \mu \right)$ from definition. By induction hypothesis, assume the above expression holds at $n = k$. For $n = k+1$,

$$V_{k+1}(x,y) := \mathbf{T}^{k+1}[V_0](x,y) = \mathbf{T}[V_k](x,y)$$

$$= \min_{a \in \mathcal{A}} \left[ C(x,a) + \gamma \max_{\xi \in \mathcal{U}_{\text{CVaR}}(y, P(\cdot|x,a))} \sum_{x' \in \mathcal{X}} \xi(x') V_k \left( x', \underbrace{y\xi(x')}_{y'} \right) P(x'|x,a) \right]$$

$$= \min_{a \in \mathcal{A}} \left[ C(x,a) + \gamma \max_{\xi \in \mathcal{U}_{\text{CVaR}}(y, P(\cdot|x,a))} \sum_{x' \in \mathcal{X}} \xi(x') P(x'|x,a) \min_{\mu \in \Pi_M} \text{CVaR}_{y'} \left( \mathcal{C}_{0,k} + \gamma^k V_0 \mid x_0 = x', \mu \right) \right]$$

$$= \min_{a \in \mathcal{A}} \left[ C(x,a) + \max_{\xi \in \mathcal{U}_{\text{CVaR}}(y, P(\cdot|x,a))} \mathbb{E}_\xi \left[ \min_{\mu \in \Pi_M} \text{CVaR}_{y_1} \left( \mathcal{C}_{1,k+1} + \gamma^{k+1} V_0 \mid x_1, \mu \right) \right] \right]$$

$$= \min_{\mu \in \Pi_M} \text{CVaR}_y \left( \mathcal{C}_{0,k+1} + \gamma^{k+1} V_0 \mid x_0 = x, \mu \right),$$

$$\tag{12}$$

where the initial state condition is given by $(x_0, y_0) = (x, y)$. Thus, the equality in (11) is proved by induction.

The second part of the proof is to show that $V^*(x_0, y_0) = \min_{\mu \in \Pi_M} \text{CVaR}_{y_0} (\lim_{n \to \infty} \mathcal{C}_{0,n} \mid x_0, \mu)$. Recall $\mathbf{T}[V](x, y) = \min_{a \in \mathcal{A}} C(x, a) + \gamma \max_{\xi \in \mathcal{U}_{\text{CVaR}}(y, P(\cdot|x,a))} \mathbb{E}_\xi[V \mid x, y, a]$. Since $\mathbf{T}$ is a contraction and $V_0$ is bounded, one obtains

$$V^*(x, y) = \mathbf{T}[V^*](x, y) = \lim_{n \to \infty} \mathbf{T}^n[V_0](x, y) = \lim_{n \to \infty} V_n(x, y)$$

for any $(x, y) \in \mathcal{X} \times \mathcal{Y}$. The first and the second equality follow directly from Proposition 2.1 and Proposition 2.2 in [3] and the third equality follows from the definition of $V_n$. Furthermore since $V_0(x, y)$ is bounded for any $(x, y) \in \mathcal{X} \times \mathcal{Y}$, the result in (12) implies

$$- \lim_{n \to \infty} \gamma^n \|V_0\|_\infty \leq V^*(x_0, y_0) - \min_{\mu \in \Pi_M} \text{CVaR}_{y_0} \left( \lim_{n \to \infty} \mathcal{C}_{0,n} \mid x_0, \mu \right) \leq \lim_{n \to \infty} \gamma^n \|V_0\|_\infty.$$

Therefore, by taking $n \to \infty$, we have just shown that for any $(x_0, y_0) \in \mathcal{X} \times \mathcal{Y}$, $V^*(x_0, y_0) = \min_{\mu \in \Pi_M} \text{CVaR}_{y_0} (\lim_{n \to \infty} \mathcal{C}_{0,n} \mid x_0, \mu)$.

The third part of the proof is to show that for the initial state $x_0$ and confidence interval $y_0$, we have that

$$V^*(x_0, y_0) = \min_{\mu \in \Pi_H} \text{CVaR}_{y_0} \left( \lim_{n \to \infty} \mathcal{C}_{0,n} \mid x_0, \mu \right).$$

At any $(x_t, y_t) \in \mathcal{X} \times \mathcal{Y}$, we first define the $t^{\text{th}}$ tail-subproblem of problem (3) as follows:

$$\mathbb{V}(x_t, y_t) = \min_{\mu \in \Pi_H} \text{CVaR}_{y_t} \left( \lim_{n \to \infty} \mathcal{C}_{t,n} \mid x_t, \mu \right)$$

where the tail policy sequence is equal to $\mu = \{\mu_t, \mu_{t+1}, \ldots\}$ and the action is given by $a_j = \mu_j(h_j)$ for $j \geq t$. For any history depend policy $\widetilde{\mu} \in \Pi_H$, we also define the $\widetilde{\mu}$−induced value function as $\text{CVaR}_{y_t} (\lim_{n \to \infty} \mathcal{C}_{t,n} \mid x_t, \widetilde{\mu})$ where $\widetilde{\mu} = \{\widetilde{\mu}_t, \widetilde{\mu}_{t+1}, \ldots\}$ and $a_j = \widetilde{\mu}_j(h_j)$ for $j \geq t$.

Now let $\mu^*$ be the optimal policy of the above $t^{\text{th}}$ tail-subproblem. Clearly, the truncated policy $\widetilde{\mu} = \{\mu_{t+1}^*, \mu_{t+2}^*, \ldots\}$ is a feasible policy for the $(t+1)^{\text{th}}$ tail subproblem at any state $x_{t+1}$ and confidence interval $y_{t+1}$:

$$\min_{\mu \in \Pi_H} \text{CVaR}_{y_{t+1}} \left( \lim_{n \to \infty} \mathcal{C}_{t+1,n} \mid x_{t+1}, \mu \right).$$

Collecting the above results, for any pair $(x_t, y_t) \in \mathcal{X} \times \mathcal{Y}$ and with $a_t = \mu_t^*(x_t)$ we can write

$$\mathbb{V}(x_t, y_t) = C(x_t, a_t) + \gamma \max_{\xi \in \mathcal{U}_{\text{CVaR}}(y_t, P(\cdot|x_t, a_t))} \mathbb{E} \left[ \xi(x_{t+1}) \cdot \underbrace{\text{CVaR}_{y_{t+1}} \left( \lim_{n \to \infty} \mathcal{C}_{t+1,n} \mid x_{t+1}, \widetilde{\mu} \right)}_{\mathbb{V}^{\widetilde{\mu}}(x_{t+1}, y_{t+1}), \, y_{t+1} = y_t \xi(x_{t+1})} \right]$$

$$\geq C(x_t, a_t) + \gamma \max_{\xi \in \mathcal{U}_{\text{CVaR}}(y_t, P(\cdot|x_t, a_t))} \mathbb{E}_\xi [\mathbb{V}(x_{t+1}, y_t \xi(x_{t+1})) \mid x_t, y_t, a_t] \geq \mathbf{T}[\mathbb{V}](x_t, y_t).$$

The first equality follows from the definition of $\mathbb{V}(x_t, y_t)$ and the decomposition of CVaRs (Theorem 2). The first inequality uses the inequality: $\mathbb{V}^{\widetilde{\mu}}(x, y) \geq \mathbb{V}(x, y), \forall (x, y) \in \mathcal{X} \times \mathcal{Y}$. The second inequality follows from the definition of Bellman operator $\mathbf{T}$.

On the other hand, starting at any state $x_{t+1}$ and confidence interval $y_{t+1}$, let $\mu^* = \{\mu_{t+1}^*, \mu_{t+2}^*, \ldots\} \in \Pi_H$ be an optimal policy for the tail subproblem:

$$\min_{\mu \in \Pi_H} \text{CVaR}_{y_{t+1}} \left( \lim_{n \to \infty} \mathcal{C}_{t+1,n} \mid x_{t+1}, \mu \right).$$

For a given pair of $(x_t, y_t) \in \mathcal{X} \times \mathcal{Y}$, construct the "extended" policy $\widetilde{\mu} = \{\widetilde{\mu}_t, \widetilde{\mu}_{t+1}, \ldots\} \in \Pi_H$ as follows:

$$\widetilde{\mu}_t(x_t) = u^*(x_t, y_t), \text{ and } \widetilde{\mu}_j(h_j) = \mu_j^*(h_j) \text{ for } j \geq t+1,$$

where $u^*(x_t, y_t)$ is the minimizer of the fixed point equation

$$u^*(x_t, y_t) \in \operatorname*{argmin}_{a \in \mathcal{A}} C(x_t, a) + \gamma \max_{\xi \in \mathcal{U}_{\text{CVaR}}(y_t, P(\cdot|x_t, a))} \mathbb{E}_\xi [\mathbb{V}(x_{t+1}, y_t \xi(x_{t+1})) \mid x_t, y_t, a],$$

with $y_t$ is the given confidence interval to the $t^{\text{th}}$ tail-subproblem and the transition from $y_t$ to $y_{t+1}$ is given by $y_{t+1} = y_t \xi^*(x_{t+1})$ where

$$\xi^* \in \arg \max_{\xi \in \mathcal{U}_{\text{CVaR}}(y_t, P(\cdot | x_t, a^*))} \mathbb{E}\left[ \xi(x_{t+1}) \cdot \text{CVaR}_{y_t \xi(x_{t+1})} \left( \lim_{n \to \infty} \mathcal{C}_{t+1,n} \mid x_{t+1}, \widetilde{\mu} \right) \right]$$

Since $\mu^*$ is an optimal, and a fortiori feasible policy for the tail subproblem (from time $t + 1$), the policy $\widetilde{\mu} \in \Pi_H$ is a feasible policy for the tail subproblem (from time $t$): $\min_{\mu \in \Pi_H} \text{CVaR}_{y_t} \left( \lim_{n \to \infty} \mathcal{C}_{t,n} \mid x_t, \mu \right)$. Hence, we can write

$$\mathbb{V}(x_t, y_t) \leq C(x_t, \widetilde{\mu}_t(x_t)) + \gamma \text{CVaR}_{y_t} \left( \lim_{n \to \infty} \mathcal{C}_{t+1,n} \mid x_t, \widetilde{\mu} \right).$$

Hence from the definition of $\mu^*$, one easily obtains:

$$\mathbb{V}(x_t, y_t)$$

$$\leq C(x_t, u^*(x_t, y_t)) + \gamma \max_{\xi \in \mathcal{U}_{\text{CVaR}}(y_t, P(\cdot | x_t, u^*(x_t, y_t)))} \mathbb{E}\left[ \xi(x_{t+1}) \cdot \text{CVaR}_{y_t \xi(x_{t+1})} \left( \lim_{n \to \infty} \mathcal{C}_{t+1,n} \mid x_{t+1}, \widetilde{\mu} \right) \mid x_t, y_t, u^*(x_t, y_t) \right]$$

$$= C(x_t, u^*(x_t, y_t)) + \gamma \max_{\xi \in \mathcal{U}_{\text{CVaR}}(y_t, P(\cdot | x_t, u^*(x_t, y_t)))} \mathbb{E}_\xi [\mathbb{V}(x_{t+1}, y_t \xi(x_{t+1})) \mid x_t, y_t, u^*(x_t, y_t)]$$

$$= \mathbf{T}[\mathbb{V}](x_t, y_t).$$

Collecting the above results, we have shown that $\mathbb{V}$ is a fixed point solution to $V(x, y) = \mathbf{T}[V](x, y)$ for any $(x, y) \in \mathcal{X} \times \mathcal{Y}$. Since the fixed point solution is unique, combining both of these arguments implies $V^*(x, y) = \mathbb{V}(x, y)$ for any $(x, y) \in \mathcal{X} \times \mathcal{Y}$. Therefore, it follows that with initial state $(x, y)$, we have $V^*(x, y) = \mathbb{V}(x, y) = \min_{\mu \in \Pi_H} \text{CVaR}_y \left( \lim_{T \to \infty} \mathcal{C}_{0,T} \mid x_0 = x, \mu \right)$.

Combining the above three parts of the proof, the claims of this theorem follows.

### A.4  Proof of Theorem 5

Similar to the definition of the optimal Bellman operator $\mathbf{T}$, for any augmented stationary Markovin policy $u : \mathcal{X} \times \mathcal{Y} \to \mathcal{A}$, we define the policy induced Bellman operator $\mathbf{T}_u$ as

$$\mathbf{T}_u[V](x, y) = C(x, u(x, y)) + \gamma \max_{\xi \in \mathcal{U}_{\text{CVaR}}(y, P(\cdot | x, u(x, y)))} \sum_{x' \in \mathcal{X}} \xi(x') V(x', y\xi(x')) P(x' | x, u(x, y)).$$

Analogous to Theorem 4, we can easily show that the fixed point solution to $\mathbf{T}_u[V](x, y) = V(x, y)$ is unique and the CVaR decomposition theorem (Theorem 2) further implies this solution equals to

$$\text{CVaR}_y \left( \lim_{T \to \infty} \mathcal{C}_{0,T} \mid x_0 = x, u_H \right),$$

where the history dependent policy $\pi_H = \{\mu_0, \mu_1, \ldots\}$ is given by $\mu_k(h_k) = u(x_k, y_k)$ for any $k \geq 0$, with initial states $x_0, y_0 = \alpha$, state transitions (8), but with augmented stationary Markovian policy $u^*$ replaced by $u$.

To complete the proof of this theorem, we need to show that the augmented stationary Markovian policy $u^*$ is optimal if and only if

$$\mathbf{T}[V^*](x, y) = \mathbf{T}_{u^*}[V^*](x, y), \ \forall x \in \mathcal{X}, \ y \in \mathcal{Y}, \tag{13}$$

where $V^*(x, y)$ is the unique fixed point solution of $\mathbf{T}[V](x, y) = V(x, y)$. Here an augmented stationary Markovian policy $u^*$ is optimal if and only if the induced history dependent policy $u_H^*$ in (7) is optimal to problem (3).

First suppose $u^*$ is an optimal augmented stationary Markvoian policy. Then using the definition of $u^*$ and the result from Theorem 4 that

$$V^*(x, y) = \min_{\mu \in \Pi_H} \text{CVaR}_y \left( \lim_{T \to \infty} \mathcal{C}_{0,T} \mid x_0 = x, \mu \right),$$

we immediately show that $V^*(x, y) = V_{u^*}(x, y)$. By the fixed point equation $\mathbf{T}[V^*](x, y) = V^*(x, y)$ and $\mathbf{T}_{u^*}[V_{u^*}](x, y) = V_{u^*}(x, y)$, this further implies (13) holds.

Second suppose $u^*$ satisfies the equality in (13). Then by the fixed point equality $\mathbf{T}[V^*](x, y) = V^*(x, y)$, we immediately obtain the equation $V^*(x, y) = \mathbf{T}_{u^*}[V^*](x, y)$ for any $x \in \mathcal{X}$ and $y \in \mathcal{Y}$. since the fixed point solution to $\mathbf{T}_{u^*}[V](x, y) = V(x, y)$

is unique, we further show that $\mathbf{T}[V^*](x,y) = V^*(x,y) = V_{u^*}(x,y)$ and $V_{u^*}(x,y) = \min_{\mu \in \Pi_H} \mathrm{CVaR}_y \left( \lim_{T \to \infty} \mathcal{C}_{0,T} \mid x_0 = x, \mu \right)$ from Theorem 4. By using the policy construction formula in (7) to obtain the history dependent policy $u_H^*$ and following the above arguments at which the augmented Markovian stationary policy $u$ is replaced by $u^*$, this further implies

$$\min_{\mu \in \Pi_H} \mathrm{CVaR}_y \left( \lim_{T \to \infty} \mathcal{C}_{0,T} \mid x_0 = x, \mu \right) = \mathrm{CVaR}_y \left( \lim_{T \to \infty} \mathcal{C}_{0,T} \mid x_0 = x, u_H^* \right),$$

i.e., $u^*$ is an optimal augmented stationary Markovian policy.

### A.5 Proof of Lemma 6

We first proof the monotonicity property. Based on the definition of $\mathcal{I}_x[V](y)$, if $V_1(x,y) \geq V_2(x,y)$ $\forall x \in \mathcal{X}$ and $y \in \mathcal{Y}$, we have that

$$\mathcal{I}_x[V_1](y) = \frac{y_{i+1} V_1(x, y_{i+1})(y - y_i) + y_i V_1(x, y_i)(y_{i+1} - y)}{y_{i+1} - y_i}, \text{ if } y \in \mathbf{I}_i(x).$$

Since $y_i, y_{i+1} \in \mathcal{Y}$ and $(y_{i+1} - y), (y - y_i) \geq 0$ (because $y \in \mathbf{I}_i(x)$), we can easily see that $\mathcal{I}_x[V_1](y) \geq \mathcal{I}_x[V_2](y)$. As $y \in \mathcal{Y}$ and $\xi(\cdot)P(\cdot|x,a) \geq 0$ for any $\xi \in \mathcal{U}_{\mathrm{CVaR}}(y, P(\cdot|x,a))$, this further implies $\mathbf{T}_{\mathcal{I}}[V_1](x,y) \geq \mathbf{T}_{\mathcal{I}}[V_2](x,y)$.

Next we prove the constant shift property. Note from the definition of $\mathcal{I}_x[V](y)$ that

$$\mathcal{I}_x[V + K](y)$$
$$= y_i(V(x,y_i) + K) + \frac{y_{i+1}(V(x,y_{i+1}) + K) - y_i(V(x,y_i) + K)}{y_{i+1} - y_i}(y - y_i), \text{ if } y \in \mathbf{I}_i(x),$$
$$= yK + y_i V(x,y_i) + \frac{y_{i+1} V(x,y_{i+1}) - y_i V(x,y_i)}{y_{i+1} - y_i}(y - y_i), \text{ if } y \in \mathbf{I}_i(x)$$
$$= \mathcal{I}_x[V](y) + yK.$$

Therefore by definition of $\mathbf{T}_{\mathcal{I}}[V](x,y)$, the constant shift property: $\mathbf{T}_{\mathcal{I}}[V + K](x,y) = \mathbf{T}_{\mathcal{I}}[V](x,y) + \gamma K$ for any $x \in \mathcal{X}$, $y \in \mathcal{Y}$, follows directly from the above arguments.

Equipped with both properties in monotonicity and constant shift, the proof of contraction of $\mathbf{T}_{\mathcal{I}}$ directly follows from the analogous proof in Lemma 3.

Finally we prove the concavity preserving property. Assume $yV(x,y)$ is concave in $y \in \mathcal{Y}$ for any $x \in \mathcal{X}$. Then for $y_{i+2} > y_{i+1} > y_i, \forall i \in \{1, \ldots, N(x) - 2\}$ the following inequality immediately follows from the definition of a concave function:

$$\frac{d\mathcal{I}_x[V](y)}{dy}\bigg|_{y \in \mathbf{I}_{i+1}(x)} = \frac{y_{i+1} V(x, y_{i+1}) - y_i V(x, y_i)}{y_{i+1} - y_i}$$
$$\geq \frac{y_{i+2} V(x, y_{i+2}) - y_{i+1} V(x, y_{i+1})}{y_{i+2} - y_{i+1}} = \frac{d\mathcal{I}_x[V](y)}{dy}\bigg|_{y \in \mathbf{I}_{i+2}(x)}. \tag{14}$$

We then show that the following inequality in each of the following cases, whenever the slope exists:

$$\mathcal{I}_x[V](z_1) \leq \mathcal{I}_x[V](z_2) + \frac{d\mathcal{I}_x[V](y)}{dy}\bigg|_{y=z_2} (z_1 - z_2), \ \forall z_1, z_2 \in \mathcal{Y} \setminus \{0\}.$$

(1) There exists $i \in \{1, \ldots, N(x) - 1\}$ such that $z_1, z_2 \in \mathbf{I}_{i+1}(x)$. In this case we have that

$$\frac{d\mathcal{I}_x[V](y)}{dy}\bigg|_{y=z_1} = \frac{d\mathcal{I}_x[V](y)}{dy}\bigg|_{y=z_2},$$

and this further implies

$$\mathcal{I}_x[V](z_1) = \mathcal{I}_x[V](z_2) + \frac{d\mathcal{I}_x[V](y)}{dy}\bigg|_{y=z_2} (z_1 - z_2).$$

(2) There exists $i, j \in \{1, \ldots, N(x) - 2\}$, $i + 1 < j$ such that $z_1 \in \mathbf{I}_{i+1}(x)$ and $z_2 \in \mathbf{I}_j(x)$. In this case, without loss of generality we assume $j = i + 1$. The proof for case: $j > i + 2$ is omitted

for the sake of brevity, as it can be completed by iteratively applying the same arguments from case: $j = i + 2$. Since $z_1 \in \mathbf{I}_i(x)$, $z_2 \in \mathbf{I}_j(x)$, we have $z_2 - z_1 \geq 0$ and

$$\left.\frac{d\mathcal{I}_x[V](y)}{dy}\right|_{y=z_1} \geq \left.\frac{d\mathcal{I}_x[V](y)}{dy}\right|_{y=z_2}.$$

Based on the definition of the linear interpolation function, we have that

$$\mathcal{I}_x[V](y_{i+1}) = y_{i+1}V(x, y_{i+1}) = \mathcal{I}_x[V](y_i) + \left.\frac{d\mathcal{I}_x[V](y)}{dy}\right|_{y \in \mathbf{I}_{i+1}(x)}(y_{i+1} - y_i).$$

Furthermore, combining previous arguments with the definitions of $\mathcal{I}_x[V](z_1)$, $\mathcal{I}_x[V](z_2)$ implies that for $(z_2 - y_{i+1}) \geq 0$,

$$\begin{aligned}
\mathcal{I}_x[V](z_2) =& \mathcal{I}_x[V](y_{i+1}) + \left.\frac{d\mathcal{I}_x[V](y)}{dy}\right|_{y=z_2}(z_2 - y_{i+1}) \\
\leq& \mathcal{I}_x[V](y_{i+1}) + \left.\frac{d\mathcal{I}_x[V](y)}{dy}\right|_{y=z_1}(z_2 - y_{i+1}) \\
=& \mathcal{I}_x[V](y_i) + \left.\frac{d\mathcal{I}_x[V](y)}{dy}\right|_{y \in \mathbf{I}_{i+1}(x)}(z_2 - y_i) \\
=& \mathcal{I}_x[V](z_1) + \left.\frac{d\mathcal{I}_x[V](y)}{dy}\right|_{y=z_1}(z_2 - z_1).
\end{aligned}$$

(3) There exists $i, j \in \{1, \ldots, N(x) - 2\}$, $i + 1 < j$ such that $z_2 \in \mathbf{I}_{i+1}(x)$ and $z_1 \in \mathbf{I}_j(x)$. In this case, without loss of generality we assume $j = i + 1$. The proof for case: $j > i + 2$ is omitted for the sake of brevity, as it can be completed by iteratively applying the same arguments from case: $j = i + 2$. Since $z_2 \in \mathbf{I}_{i+1}(x)$, $z_1 \in \mathbf{I}_j(x)$, we have $z_1 - z_2 \geq 0$ and

$$\left.\frac{d\mathcal{I}_x[V](y)}{dy}\right|_{y=z_1} \leq \left.\frac{d\mathcal{I}_x[V](y)}{dy}\right|_{y=z_2}.$$

Similar to the analysis in the previous case, we have that

$$\mathcal{I}_x[V](y_i) = y_i V(x, y_i) = \mathcal{I}_x[V](y_{i+1}) + \left.\frac{d\mathcal{I}_x[V](y)}{dy}\right|_{y \in \mathbf{I}_{i+1}(x)}(y_i - y_{i+1})$$

Furthermore, combining previous arguments with the definitions of $\mathcal{I}_x[V](z_1)$, $\mathcal{I}_x[V](z_2)$ implies that for $(z_2 - z_1) \leq 0$,

$$\begin{aligned}
\mathcal{I}_x[V](z_2) =& \mathcal{I}_x[V](y_i) + \left.\frac{d\mathcal{I}_x[V](y)}{dy}\right|_{y=z_2}(z_2 - y_i) \\
=& \mathcal{I}_x[V](y_{i+1}) + \left.\frac{d\mathcal{I}_x[V](y)}{dy}\right|_{y=z_2}(z_2 - y_{i+1}) \\
=& \mathcal{I}_x[V](z_1) + \left.\frac{d\mathcal{I}_x[V](y)}{dy}\right|_{y=z_2}(z_2 - z_1) \\
\leq& \mathcal{I}_x[V](z_1) + \left.\frac{d\mathcal{I}_x[V](y)}{dy}\right|_{y=z_1}(z_2 - z_1).
\end{aligned}$$

Thus we have just shown that the first order sufficient condition for concave functions, corresponding to $\mathcal{I}_x[V](y)$, holds, i.e., $\mathcal{I}_x[V](y)$ is concave in $y \in \mathcal{Y} \setminus \{0\}$ for any given $x \in \mathcal{X}$. Now since $\mathcal{I}_x[V](y)$ is a continuous piecewise linear function in $y \in \mathcal{Y}$ and a concave function when the domain is restricted to $\mathcal{Y} \setminus \{0\}$. By continuity this immediately implies that $\mathcal{I}_x[V](y)$ is concave in $y \in \mathcal{Y}$ as well. Then following the identical arguments in the proof of Lemma 3 for the concavity preserving property, we can thereby show that

$$y\mathbf{T}_\mathcal{I}[V](x, y) = \min_{a \in \mathcal{A}} \left\{ yC(x, a) + \max_{\xi \in \mathcal{U}_{\text{CVaR}}(y, P(\cdot|x,a))} \sum_{x' \in \mathcal{X}} \mathcal{I}_{x'}[V](y\xi(x'))P(x'|x, a) \right\}$$

is concave in $y \in \mathcal{Y}$ for any given $x \in \mathcal{X}$.

## A.6 Useful Intermediate Results

**Lemma 8** *Let $f(y) : [0,1] \to R$ be a concave function, differentiable almost everywhere, with Lipschitz constant $M$. Then the linear interpolation $\mathcal{I}[f](y)$ is also concave, and with Lipschitz constant $M_I \leq M$.*

**Proof**   For every segment $[y_j, y_{j+1}]$ in the linear interpolation, $f(y)$ is concave, and with Lipschitz constant $M$, and $\mathcal{I}[f](y)$ is linear. Also, $f(y_j) = \mathcal{I}[f](y_j)$, and $f(y_{j+1}) = \mathcal{I}[f](y_{j+1})$, by definition of the linear interpolation. Denote by $c_j$ the magnitude of the slope of $\mathcal{I}[f](y)$ at $y \in [y_j, y_{j+1}]$.

Assume by contradiction that $c_j > \max_{y \in [y_j, y_{j+1}]} |f'(y)|$ whenever $f'(y)$ exists. Consider the case when $f(y_{j+1}) \geq f(y_j)$. This implies $c_j$ is the slope of the interpolation function $\mathcal{I}[f](y)$ at $y \in [y_j, y_{j+1}]$. Then by the fundamental theorem of calculus, we have

$$f(y_{j+1}) - f(y_j) = \int_{y_j}^{y_{j+1}} f'(y)dy \leq \int_{y_j}^{y_{j+1}} |f'(y)|dy < \int_{y_j}^{y_{j+1}} c_j dy = (\mathcal{I}[f](y_{j+1}) - \mathcal{I}[f](y_j)),$$

contradicting $f(y_{j+1}) = \mathcal{I}[f](y_{j+1})$ and $f(y_j) = \mathcal{I}[f](y_j)$.

On the other hand, consider the case when $f(y_{j+1}) \leq f(y_j)$. This implies $-c_j$ is the slope of the interpolation function $\mathcal{I}[f](y)$ at $y \in [y_j, y_{j+1}]$. Again by fundamental theorem of calculus,

$$0 \leq f(y_{j+1}) - f(y_j) = \int_{y_j}^{y_{j+1}} f'(y)dy \geq \int_{y_j}^{y_{j+1}} -|f'(y)|dy > \int_{y_j}^{y_{j+1}} -c_j dy = \mathcal{I}[f](y_j) - \mathcal{I}[f](y_{j+1}).$$

Since $f(y_{j+1}) = \mathcal{I}[f](y_{j+1})$ and $f(y_j) = \mathcal{I}[f](y_j)$, which implies $\mathcal{I}[f](y_j) - \mathcal{I}[f](y_{j+1}) \geq 0$, the above expression clearly leads to a contradiction.

We finally have that $\max_{y \in [y_j, y_{j+1}]} |f'(y)| \geq c_j$ for segment $j \in \{1, \ldots, N(x) - 1\}$. As this argument holds for each segment, by maximizing over $j$ over $\{1, \ldots, N(x) - 1\}$, we have that

$$M \geq \max_{j \in \{1,\ldots,N(x)-1\}} \max_{y \in [y_j, y_{j+1}]} |f'(y)| \geq \max_{j \in \{1,\ldots,N(x)-1\}} c_j = M_I.$$

The concavity property (thus differentiability almost everywhere) are well-known results of linear interpolation [18].

**Lemma 9** *Let $yV(x, y)$ be Lipschitz with constant $M$, concave, and differentiable almost everywhere, for every $x \in \mathcal{X}$ and $y \in [0, 1]$. Then $y\mathbf{T}[V](x, y)$ is also Lipschitz with constant $C_{\max} + \gamma M$.*

**Proof**   For any given state-action pair $x \in \mathcal{X}$, and $a \in \mathcal{A}$, let $P(x') = P(x'|x, a)$ be the transition kernel. Consider the function

$$H(y) \doteq \max_{\xi \in \mathcal{U}_{\mathrm{CVaR}}(y, P(\cdot))} \sum_{x' \in \mathcal{X}} y\xi(x')V(x', y\xi(x'))P(x').$$

Note that, by definition of $\mathcal{U}_{\mathrm{CVaR}}$, and a change of variables $z(x') = y\xi(x')$, we can write $H(y)$ as follows:

$$H(y) = \max_{\substack{0 \leq z(x') \leq 1, \\ \sum_{x'} P(x')z(x') = y}} \sum_{x' \in \mathcal{X}} z(x')V(x', z(x'))P(x'). \tag{15}$$

The Lagrangian of the above maximization problem is

$$L(z, \lambda; y) = \sum_{x' \in \mathcal{X}} z(x')V(x', z(x'))P(x') - \lambda(\sum_{x'} P(x')z(x') - y).$$

Since $yV(x, y)$ is concave, the maximum is attained. By first order optimality condition the following expression holds:

$$\frac{\partial L(z, \lambda; y)}{\partial z(x')} = P(x')\frac{\partial [z(x')V(x', z(x'))]}{\partial z(x')} - \lambda P(x') = 0.$$

Summing the last expression over $x'$, we obtain:

$$\sum_{x' \in \mathcal{X}} P(x')\frac{\partial [z(x')V(x', z(x'))]}{\partial z(x')} = \sum_{x' \in \mathcal{X}} \lambda P(x') = \lambda.$$

Now, from the Lipschitz property of $yV(x, y)$, we have

$$\left| \sum_{x' \in \mathcal{X}} \lambda P(x') \right| \leq \sum_{x' \in \mathcal{X}} P(x') \left| \frac{\partial \left[ z(x')V\left(x', z(x')\right) \right]}{\partial z(x')} \right| \leq \sum_{x' \in \mathcal{X}} P(x')M = M.$$

Thus,

$$|\lambda| \leq \sum_{x' \in \mathcal{X}} P(x') \left| \frac{\partial \left[ z(x')V\left(x', z(x')\right) \right]}{\partial z(x')} \right| \leq M.$$

Note that the objective in (15) does not depend on $y$. From the envelope theorem [14], it follows that

$$\frac{dH(y)}{dy} = \lambda,$$

therefore, $H(y)$ is Lipschitz, with constant $M$.

Now, by definition,

$$y\mathbf{T}[V](x, y) = \min_{a \in \mathcal{A}} \left[ yC(x, a) + \gamma \max_{\xi \in \mathcal{U}_{\text{CVaR}}(y, P(\cdot|x, a))} \sum_{x' \in \mathcal{X}} y\xi(x')V\left(x', y\xi(x')\right) P(x'|x, a) \right].$$

Using our Lipschitz result for $H(y)$, we have that for any $a \in \mathcal{A}$, the function

$$yC(x, a) + \gamma \max_{\xi \in \mathcal{U}_{\text{CVaR}}(y, P(\cdot|x, a))} \sum_{x' \in \mathcal{X}} y\xi(x')V\left(x', y\xi(x')\right) P(x'|x, a)$$

is Lipschitz in $y$, with constant $C(x, a) + \gamma M$. Using again the envelope theorem [14], we obtain that $y\mathbf{T}[V](x, y)$ is Lipschitz, with constant $C_{\max} + \gamma M$.

**Lemma 10** *Consider Algorithm 1. Assume that for any $x \in \mathcal{X}$, the initial value function satisfies that $yV_0(x, y)$ is Lipschitz (in $y$), with uniform constant $M_0$. We have that for any $t \in \{0, 1, \ldots, \}$, the function $yV_t(x, y)$ is Lipschitz in $y$ for any $x \in \mathcal{X}$, with Lipschitz constant*

$$M_t = \frac{1 - \gamma^t}{1 - \gamma}C_{\max} + \gamma^t M_0 \leq \frac{C_{\max}}{1 - \gamma} + M_0, \ \forall t.$$

**Proof** Let $\mathbf{T}_{\mathcal{I}}[V]$ denote the application of the Bellman operator $\mathbf{T}$ to the linearly-interpolated version of $yV(x, y)$. We have, by definition, that

$$V_1(x, y) = \mathbf{T}_{\mathcal{I}}[V_0](x, y).$$

Using Lemma 8 and Lemma 9, we have that $V_1(x, y)$ is Lipschitz, with $M_1 \leq C_{\max} + \gamma M_0$.

Note now, that $V_2(x, y) = \mathbf{T}_{\mathcal{I}}[V_1](x, y)$. Thus, by induction, we have

$$M_t \leq \frac{1 - \gamma^t}{1 - \gamma}C_{\max} + \gamma^t M_0,$$

and the result follows.

### A.7 Proof of Theorem 7

The proof of this theorem is split into three parts. In the first part, we bound the difference $\mathcal{I}_x[V_t](y)/y - V_t(x, y)$ at each state $(x, y) \in \mathcal{X} \times \mathcal{Y}$ using the previous technical lemmas and Lipschitz property.

In the second part, we bound the difference of $\mathbf{T}_{\mathcal{I}}[V_t](x, y) - \mathbf{T}[V_t](x, y)$.

In the third part we bound the interpolation error using contraction properties of Bellman recursions.

First we analyze the bounds for $\mathcal{I}_x[V_t](y)/y - V_t(x, y)$ in the following four cases. Notice that from Lemma 10, we have that $|d\mathcal{I}_x[V_t](y)/dy| \leq M := C_{\max}/(1 - \gamma) + M_0$.

(1) When $y = 0$ (for which $y \in \mathbf{I}_1(x)$).
Using previous analysis and L'Hospital's rule we have that $\lim_{y \to 0} \mathcal{I}_x[V_t](y)/y = V_t(x, 0)$. This further implies $\lim_{y \to 0} \mathcal{I}_x[V_t](y)/y - V_t(x, 0) = 0$.

(2) When $y \in \mathbf{I}_{i+1}(x)$, $2 \le i < N(x) - 1$.

Similar to the inequality in (14), by concavity of $yV_t(x, y)$ in $y \in \mathcal{Y}$, we have that

$$\left. \frac{d\mathcal{I}_x[V_t](y)}{dy} \right|_{y \in \mathbf{I}_{i+1}(x)} = \frac{y_{i+1}V_t(x, y_{i+1}) - y_i V_t(x, y_i)}{y_{i+1} - y_i} \le \frac{yV_t(x, y) - y_i V_t(x, y_i)}{y - y_i},$$

and

$$\left. \frac{d\mathcal{I}_x[V_t](y)}{dy} \right|_{y \in \mathbf{I}_{i+2}(x)} = \frac{y_{i+2}V_t(x, y_{i+2}) - y_{i+1}V_t(x, y_{i+1})}{y_{i+2} - y_{i+1}} \le \frac{y_{i+1}V_t(x, y_{i+1}) - yV_t(x, y)}{y_{i+1} - y}.$$

From the first inequality, for each $(x, y) \in \mathcal{X} \times \mathcal{Y}$ we get,

$$\frac{\mathcal{I}_x[V_t](y)}{y} - V_t(x, y) \le \frac{1}{y} \left( y_i V_t(x, y_i) + \frac{y_{i+1}V_t(x, y_{i+1}) - y_i V_t(x, y_i)}{y_{i+1} - y_i}(y - y_i) - yV_t(x, y) \right) \le 0. \tag{16}$$

On the other hand, rearranging the second inequality gives

$$\frac{1}{y}(\mathcal{I}_x[V_t](y) - yV_t(x, y))$$

$$\ge \frac{1}{y} \left( y_i V_t(x, y_i) + \left. \frac{d\mathcal{I}_x[V_t](y)}{dy} \right|_{y \in \mathbf{I}_{i+1}(x)}(y - y_i) - y_{i+1}V_t(x, y_{i+1}) - \left. \frac{d\mathcal{I}_x[V_t](y)}{dy} \right|_{y \in \mathbf{I}_{i+2}(x)}(y - y_{i+1}) \right)$$

$$= \left( \left. \frac{d\mathcal{I}_x[V_t](y)}{dy} \right|_{y \in \mathbf{I}_{i+1}(x)} - \left. \frac{d\mathcal{I}_x[V_t](y)}{dy} \right|_{y \in \mathbf{I}_{i+2}(x)} \right) \frac{y - y_{i+1}}{y} \ge -2M \left( \frac{y_{i+1}}{y} - 1 \right). \tag{17}$$

Furthermore by the Lipschitz property, we also have the following inequality as well:

$$\frac{1}{y}(\mathcal{I}_x[V_t](y) - yV_t(x, y))$$

$$= \frac{y_{i+1}V_t(x, y_{i+1})(y - y_i) + y_i V_t(x, y_i)(y_{i+1} - y)}{(y_{i+1} - y_i)y} - V_t(x, y)$$

$$\ge \frac{y_i V_t(x, y_i)(y - y_i) + y_i V_t(x, y_i)(y_{i+1} - y) - M(y_{i+1} - y_i)(y - y_i)}{(y_{i+1} - y_i)y} - V_t(x, y) \tag{18}$$

$$= \frac{y_i V_t(x, y_i) - M(y - y_i)}{y} - V_t(x, y) \ge -2M \left( 1 - \frac{y_i}{y} \right).$$

Combining the inequalities (17) and (18), the following lower bound for $\mathcal{I}_x[V_t](y)/y - V_t(x, y)$ holds:

$$\frac{1}{y}(\mathcal{I}_x[V_t](y) - yV_t(x, y)) \ge \delta := -2M \min \left\{ 1 - \frac{y_i}{y}, \frac{y_{i+1}}{y} - 1 \right\}, \ \forall y \in \mathbf{I}_{i+1}(x), \ i \ge 2.$$

From the above definition, when $y_i \le y \le (y_i + y_{i+1})/2$, the lower bound becomes $\delta = -2M(1 - y_i/y)$ and when $(y_i + y_{i+1})/2 \le y \le y_{i+1}$, the corresponding lower bound is $\delta = -2M(y_{i+1}/y - 1)$. In both cases, $\delta$ is minimized when $y = (y_i + y_{i+1})/2$. Therefore, the above analysis implies the following lower bound:

$$\frac{1}{y}(\mathcal{I}_x[V_t](y) - yV_t(x, y)) \ge -2M \frac{y_{i+1} - y_i}{y_{i+1} + y_i}, \ \forall y \in \mathbf{I}_{i+1}(x), \ i \ge 2.$$

When $y_{i+1} = \theta y_i$ for $i \in \{2, \ldots, N(x) - 1\}$ for some constant $\theta \ge 1$, this further implies that

$$\frac{1}{y}(\mathcal{I}_x[V_t](y) - yV_t(x, y)) \ge -2M \frac{\theta - 1}{\theta + 1} \ge -M(\theta - 1), \ \forall y \in \mathcal{Y} \setminus [0, \epsilon].$$

Then combining the results, here we get the following bound for $\mathcal{I}_x[V_t](y)/y - V_t(x, y)$:

$$-M(\theta - 1) \le \frac{\mathcal{I}_x[V_t](y)}{y} - V_t(x, y) \le 0, \ \forall y \in \mathbf{I}_{i+1}(x), \ i \ge 2.$$

(3) When $y \in \mathbf{I}_{N(x)}(x)$, i.e., $y \in (y_{N(x)-1}, 1]$.

Similar to the proof of case (2), we can show that for any $x \in \mathcal{X}$ and $y \in \mathbf{I}_{N(x)}(x)$, the same lines of arguments in inequality (16) and (18) hold, which implies

$$-2M\left(1 - y_{N(x)-1}\right) \le -2M\left(1 - \frac{y_{N(x)-1}}{y}\right) \le \frac{1}{y}(\mathcal{I}_x[V_t](y) - yV_t(x,y)) \le 0.$$

When $y_{N(x)} = 1 = \theta y_{N(x)-1}$, this further shows that

$$-2My_{N(x)-1}(\theta - 1) = -2M\left(y_{N(x)} - y_{N(x)-1}\right) \le \frac{1}{y}(\mathcal{I}_x[V_t](y) - yV_t(x,y)) \le 0,$$

and

$$-2M(\theta - 1) \le -\frac{2}{\theta}M(\theta - 1) \le \frac{1}{y}(\mathcal{I}_x[V_t](y) - yV_t(x,y)) \le 0.$$

(4) When $y \in \mathbf{I}_2(x)$, i.e., $y \in (0, y_2]$.

From inequality (16), the definition of $\mathcal{I}_x[V_t](y)$, we have that

$$0 \ge \frac{\mathcal{I}_x[V_t](y) - yV_t(x,y)}{y} = \frac{y(V_t(x,y_2) - V_t(x,y))}{y} = V_t(x,y_2) - V_t(x,y) \ge V_t(x,y_2) - V_t(x,0).$$

The first inequality is due to the fact that $yV_t(x,y)$ is concave in $y \in \mathcal{Y}$ for any $x \in \mathcal{X}$, thus the first order condition implies

$$\frac{y_2 V_n(x,y_2) - y_1 V_n(x,y_1)}{y_2 - y_1} \le \frac{yV_n(x,y) - y_1 V_n(x,y_1)}{y - y_1}, \ \forall y \in \mathbf{I}_2(x),$$

and the last inequality is due to the similar fact that

$$V_t(x,w) = \frac{wV_t(x,w) - 0 \cdot V_t(x,0)}{w - 0} \le \frac{zV_t(x,z) - 0 \cdot V_t(x,0)}{z - 0} = V_t(x,z), \ \forall z, w \in \mathcal{Y}, \ z \le w.$$

Therefore the condition of this theorem implies

$$0 \ge \frac{\mathcal{I}_x[V_t](y) - yV_t(x,y)}{y} \ge -\epsilon, \ \forall t \ge 0, \ x \in \mathcal{X}, \ y \in \mathcal{Y}.$$

Combining the above four cases, we have that for each state $(x,y) \in \mathcal{X} \times \mathcal{Y}$,

$$0 \ge \frac{\mathcal{I}_x[V_t](y)}{y} - V_t(x,y) \ge -2M(\theta - 1) - \epsilon, \ \forall t.$$

Second, we bound the difference of $\mathbf{T}_{\mathcal{I}}[V_t](x,y) - \mathbf{T}[V_t](x,y)$. By recalling that $\xi(\cdot)P(\cdot|x,a)$ is a probability distribution for any $\xi \in \mathcal{U}_{\mathrm{CVaR}}(y, P(\cdot|x,a))$, we then combine all previous arguments and show that at any $t \in \{0,1,\dots,\}$ and any $x \in \mathcal{X}$, $a \in \mathcal{A}$, $y \in \mathbf{Y}(x)$,

$$\max_{\xi \in \mathcal{U}_{\mathrm{CVaR}}(y,P(\cdot|x,a))} \sum_{x' \in \mathcal{X}, \xi(x') \ne 0} \left(\frac{\mathcal{I}_{x'}[V_t](y\xi(x'))}{y\xi(x')} - V_t(x', y\xi(x'))\right) \xi(x')P(x'|x,a) \ge -2M(\theta - 1) - \epsilon.$$

This further implies

$$\mathbf{T}[V_t](x,y) - \gamma(2M(\theta - 1) + \epsilon) \le \mathbf{T}_{\mathcal{I}}[V_t](x,y) \le \mathbf{T}[V_t](x,y). \tag{19}$$

Third, we prove the error bound of interpolation based value iteration using the above properties. By putting $t = 0$ in (19), we have that

$$-\gamma(2M(\theta - 1) + \epsilon) \le \mathbf{T}_{\mathcal{I}}[V_0](x,y) - \mathbf{T}[V_0](x,y) \le 0.$$

Applying the Bellman operator $\mathbf{T}$ on all sides of the above inequality and noting that $\mathbf{T}$ is a translational invariant mapping, the above expression implies

$$\mathbf{T}^2[V_0](x,y) - \gamma^2(2M(\theta - 1) + \epsilon) \le \mathbf{T}[\mathbf{T}_{\mathcal{I}}[V_0]](x,y) = \mathbf{T}[V_1](x,y) \le \mathbf{T}^2[V_0](x,y).$$

By adding the inequality: $-\gamma(2M(\theta - 1) + \epsilon) \le \mathbf{T}_{\mathcal{I}}[V_1](x,y) - \mathbf{T}[V_1](x,y) \le 0$ to the above expression, this further implies the following expression:

$$\mathbf{T}^2[V_0](x,y) - \gamma(1 + \gamma)(2M(\theta - 1) + \epsilon) \le \mathbf{T}_{\mathcal{I}}[V_1](x,y) = \mathbf{T}_{\mathcal{I}}^2[V_0](x,y) \le \mathbf{T}^2[V_0](x,y).$$

Figure 2: Grid-world - trajectory plots.

Then, by repeating this process, we can show that for any $n \in \mathbb{N}$, the following inequality holds:

$$\mathbf{T}^n[V_0](x,y) - \gamma \frac{1-\gamma^n}{1-\gamma}(2M(\theta-1) + \epsilon) \leq \mathbf{T}^n_{\mathcal{I}}[V_0](x,y) \leq \mathbf{T}^n[V_0](x,y).$$

Note that when $n \to \infty$, we have that $\gamma^n$ converges to 0, $\mathbf{T}^n[V_0](x,y)$ converges to $\min_{\mu \in \Pi_H} \text{CVaR}_y (\lim_{T\to\infty} \mathcal{C}_{0,T} \mid x, \mu)$ (follow from Theorem 4) and $\mathbf{T}^n_{\mathcal{I}}[V_0](x,y)$ converges to $\widehat{V}^*(x,y)$ (follow from the contraction property in Lemma 6).

Furthermore, from Proposition 1.6.4 in [3], the contraction property of Bellman operator $\mathbf{T}$ implies that for any $x \in \mathcal{X}$, $y \in \mathcal{Y}$, the following expression holds:

$$|\mathbf{T}^n[V_0](x,y) - V^*(x,y)| \leq \frac{\gamma^n}{1-\gamma}(C_{\max} + \|Z\|_\infty)$$

where $Z$ is the bounded random variable of the initial value function $V_0(x,y) = \text{CVaR}_y(Z \mid x_0 = x)$ such that $\|V_0\|_\infty \leq \|Z\|_\infty$, and $V^*(x,y) = \min_{\mu \in \Pi_H} \text{CVaR}_y (\lim_{T\to\infty} \mathcal{C}_{0,T} \mid x, \mu)$. This further implies for any $x \in \mathcal{X}$, $y \in \mathcal{Y}$,

$$|\mathbf{T}^n_{\mathcal{I}}[V_0](x,y) - V^*(x,y)| \leq \gamma \frac{1-\gamma^n}{1-\gamma}(2M(\theta-1) + \epsilon) + \frac{\gamma^n}{1-\gamma}(C_{\max} + \|Z\|_\infty).$$

Then, by combining all the above arguments, we prove the claim of this theorem.

## B   Trajectory Plots

In Figure 2 we demonstrate simulated trajectories according to a policy that is greedy w.r.t. the value function, according to Theorem 5.

## C   Generalization to Mean-CVaR Optimization

In this section we extend our approach to MDPs with a mean-CVaR objective of the form:

$$\min_{\mu \in \Pi_H} \quad \lambda \mathbb{E}\left(\lim_{T\to\infty} \mathcal{C}_{0,T} \mid x_0, \mu\right) + (1-\lambda)\text{CVaR}_\alpha\left(\lim_{T\to\infty} \mathcal{C}_{0,T} \mid x_0, \mu\right), \tag{20}$$

where $\lambda \in [0,1]$. Such an objective is common in practice [11], and is also useful for solving CVaR-constrained objectives using standard Lagrangian methods (see, e.g., [5]).

Now for any $\alpha_1, \alpha_2 \in [0,1]$, define

$$\rho_{\bar\alpha}(Z \mid H_t, \mu) = \lambda \text{CVaR}_{\alpha_1}(Z \mid H_t, \mu) + (1-\lambda)\text{CVaR}_{\alpha_2}(Z \mid H_t, \mu)$$

and notice that $\rho_{\bar\alpha}(Z \mid H_t, \mu) = \lambda \mathbb{E}(Z \mid H_t, \mu) + (1-\lambda)\text{CVaR}_\alpha(Z \mid H_t, \mu)$ when the vector of CVaR confidence intervals is given by $\bar\alpha = (1, \alpha)$.

**Theorem 11** *For any $t \geq 0$, denote by $Z \doteq (Z_{t+1}, Z_{t+2}, \dots)$ the cost sequence from time $t + 1$ onwards. The conditional mean-CVaR risk metric under policy $\mu$ obeys the following decomposition:*

$$\rho_{\bar{\alpha}}(Z \mid H_t, \mu) = \max_{\xi \in \mathcal{U}_{2CVaR}(\bar{\alpha}, P(\cdot \mid x_t, a_t))} \mathbb{E}[\mathbf{S}_\lambda(\xi(x_{t+1})) \cdot \rho_{\bar{\alpha}\mathbf{S}_\lambda(\xi(x_{t+1}))}(Z \mid H_{t+1}) \mid H_t]$$

*where $\bar{\alpha} = (\alpha_1, \alpha_2)$ is the vector of CVaR confidence intervals. The risk envelope is given by*

$$\mathcal{U}_{2CVaR}(\bar{\alpha}, P(\cdot \mid x_t, a_t)) = \left\{ \xi = (\xi_1, \xi_2) : \xi_i(x_{t+1}) \in \left[0, \frac{1}{\alpha_i}\right], \sum_{x_{t+1} \in \mathcal{X}} \xi_i(x_{t+1}) P(x_{t+1} \mid x_t, a_t) = 1, \forall i \right\},$$

$\mathbf{S}_\lambda(\xi) : \mathbb{R}^2 \mapsto \mathbb{R}$ *is a linear operator given by $\lambda \xi_1 + (1 - \lambda)\xi_2$ and $a_t$ is the control input induced by policy $\mu_t(h_t)$.*

Now we extend the above analysis to Bellman recursion. With the generic state space $\mathcal{Y} = [0, 1]^2$, we now define the optimal Bellman operator at any $(x, y) \in \mathcal{X} \times \mathcal{Y}$,

$$\mathbf{T}[V](x, y) = \min_{a \in \mathcal{A}} \left[ C(x, a) + \gamma \max_{\xi \in \mathcal{U}_{2CVaR}(y, P(\cdot \mid x, a))} \sum_{x' \in \mathcal{X}} \mathbf{S}_\lambda(\xi(x'))V\left(x', y\mathbf{S}_\lambda(\xi(x'))\right) P(x' \mid x, a) \right]. \tag{21}$$

Based on the decomposition result from Theorem 11, we now have the result on the convergence of Bellman recursion, analogous to Theorem 4 and 5, showing that the fixed point solution of $\mathbf{T}[V](x, y) = V(x, y)$ is unique and equals to the solution of (20) with $x_0 = x$ and $y_0 = (1, \alpha)$.

**Theorem 12** *For any state $x \in \mathcal{X}$ and $y = (y_1, y_2) \in [0, 1]^2$, the fixed point solution of $\mathbf{T}[V](x, y) = V(x, y)$ is unique and is equal to $V(x, y) := \min_{\mu \in \Pi_H} \lambda CVaR_{y_1}\left(\lim_{T \to \infty} \mathcal{C}_{0,T} \mid x_0, \mu\right) + (1 - \lambda)CVaR_{y_2}\left(\lim_{T \to \infty} \mathcal{C}_{0,T} \mid x_0, \mu\right)$. Furthermore, let $\mu^* = \{\mu_0, \mu_1, \dots\} \in \Pi_H$ be a policy recursively defined as in (7) with two-dimensional augmented state $\{y_j\}$ and initial condition $y_0 = (1, \alpha)$. Then $\mu^*$ is an optimal policy for the mean-CVaR problem (20) with initial condition $x_0$ and CVaR confidence level $\alpha$.*

Extending the interpolation-based CVaR value iteration (Algorithm 1) for this case is straightforward, using a 2-D linear interpolation for $yV(x, y)$.