[Reviews · NeurIPS 2015]

Submitted by Assigned_Reviewer_1

This paper addresses the problem of risk-aversive decision making within a MDP framework and CVaR. Firstly, it is showed that a CVaR measure can be interpreted as worst-case expected cost under MDP parameter perturbations given a fixed error budget, and this result provides a unified framework for risk-sensitive and robust decision making. An approximate value iteration algorithm for CVaR MDPs and its convergence rate is also presented. The approach is empirically illustrated on the problem of getting to a goal state while avoiding obstacles in 2D grid-world.

Quality: The paper is well written and technically sound. All relevant references are given.

Clarity: The paper is rather clear, but I have minor concerns in some mathematical equations as follows:

- In Line 95, equation (1), If I understood correctly, the CVaR at confidence level $\alpha$ would be defined as $CVaR_\alpha(Z) = \min_{w\in\mathbb{R}} \{ w + \frac{1}{1-\alpha} \mathbb{E}[(Z-w)^{+}]\} if $CVaR_\alpha(Z)=\mathbb{E}[Z|Z \ge VaR_\alpha(Z)]$, which is nondecreasing in $\alpha$. In the case of

$CVaR_\alpha(Z)$ decreasing in $\alpha$, (1) is correct but $CVaR_\alpha(Z)$ is written as $\mathbb E[Z|Z\ge VaR_{1-\apha}(Z)]$. - In Line 121, the space of histories up to time $t$ should be defined recursively as $H_t = H_{t-1} \times \mathcal X \times \mathcal A$. - In Line 123, deterministic history-dependent policies... - In Line 204, Theorem 2, I couldn't find Theorem 10 in [17]. Also, if I understand correctly, $H_t$ would be $h_t$ since the history up to timestep $t$ is assumed to be given. - In Line 309, it would be good if $y_{i+1}$ could be explicitly defined though its definition is quite clear from the context.

- In Line 357, $y_1$ would be written as 0 according to the assumption made in Algorithm 1. - In Line 353~359, the adaptive procedure for additional interpolation points is understandable but a bit confusing. We already have a set of interpolation points $\mathbf Y(x)=\{y_1, y_2, ..., y_{N(x)}\}$ which satisfy the condition $\log \theta = \log y_{i+1} - \log y_i$. The additional points in $(y'_2, y_2)$ are not explicitly denoted so they don't seem to be related to the condition $\log \theta \ge

\log y_{i+1} - \log y_i$. Some other notations might be needed for the additional interpolation points. If we can add new interpolation points anywhere, this addition can affect $V_t(x,y_i)$? Is $V_t(x,y_i)$ linearly interpolated rather than exactly updated from (possibly approximated) $V_{t+1}$? - In Algorithm 1, it would be good to see the list of input parameters. - The parameters used for the experiment would be mentioned (e.g. $\theta$ and $\epsilon$).

In Line 393, it is stated that $M$ is chosen to be $2/(1-\gamma)$. It would be great if the authors could explain in more detail how $M$ was chosen and it affects the computational complexity.

Originality: CVaR MDPs are not new, but this paper provides a novel equivalence between a CVaR objective and a robust MDP formulation. This is an improvement over the previously suggested equivalence in terms of conservativeness in parameter perturbations.

The paper also proposes an exact value iteration for CVaR MDPs, which runs on an MDP augmented with a continuous state variable which denotes the confidence level allowed. This challenge has been tackled by exploiting linear interpolation and the concavity, and the resulting approximate algorithm is guaranteed to converge and its error is also theoretically bounded. This paper seems sufficiently novel.

Significance: Risk-aversive planning is an interesting problem and this paper provides significant contributions.
Summary: This paper presents an equivalence between RMDPs and CVaR MDPs, and proposes a value iteration algorithm for CVaR MDPs with a convergence guarantee and a bound on approximation error. The algorithm and experiments are sounds.

Submitted by Assigned_Reviewer_2

The paper shows that a CVaR objective has an alternative interpretation as expected cost under worst-case modeling errors, if modeling errors are bounded by a budjet, bounding their product.

The paper also introduces an approximate value iteration algorithm for CVaR MDPs, which is to the best of their knowledge the first solution to CVaR MDPs with convergence error guarantees, and the first algorithm to approximate globally optimal policies for CVaR MDPs. Points that require improvement are:

(1) While proofs can be found in the supplementary material, it

seems necessary to have at least proof sketches inside the paper,

especially given that one of the two main contributions of the paper

is the proof of proposition 1.

(2) The paper heavily relies on an error budget constraint that

seems non-intuitive/non-realistic to me. It would be good to

explain whether this error-budget just makes the math easier, or

does it also have a realistic interpretation.

(3) Experiment with a more complex domain could strenghten the p

The paper is of high-quality, and very well-presented.

The paper provides significant insight on the connection between risk-sensitivity and robustness, as well as a practical algorithm for discrete state/action CVAR MDPs. Robust MDPs are an important research problem, and the paper makes progress towards better understanding and solving them.

Detailed comments by order of appearance: ========================================= - line 39 'worst-case scenario' - what does it mean: worst-case cost under worst-case parameters? or expected-cost under worst-case parameters?

- line 47 - why purturbing only transition probabilities, and not reward as well? would your result hold with reward purturbation? Some clarification is needed here, and possibly in the abstract.

- line 79 - 'compute globally optimal' - since your algorithm is an approximate VI, wouldn't it be more accurate to say "approximate globally optimal..." and mention the dependence on interpolation resolution?

- line 119 'our results easily generalize' - if they indeed generalize, it would be good to briefly explain how at a later section of the paper.

- line 122 it should be H x A x X

- line 128 you use Z_t for C(x,a) - please briefly clarify the connection explicitely

- line 144 'worst-case expected ... in presence of worst-case' => 'expected ... in presence of worst-case' Also, could you relate 'expected reward under worst-case purturbation'

to 'worst-case reward under worst-case purturbation' and possibly to

to 'worst-case reward under expected purturbation'?

- line 148 - (x_0...x_T) - it's not clear if you include actions in the trajectory, and if not, why not (later it becomes clearer when you use a deterministic, fixed policy)

- line 151 - why (x_0...x_T) starting from x_1, especially given that cost is C(x,a)?

- line 165 - the error budget seems uninutitive to me. Beyond the fact that it's not clear why errors should be bounded by a budget, this specific form of budget which bounds the product of errors seem uninutitive.

Other than helping the math, does this come in realistic contexts? if yes, could you demonstrate?

Also does your algorithm needs to know the budget (\eta) in advance? if not, what happens when: the assumed budget is smaller than the actual errors? if yes, is there a way to find a budget (upper-bound) efficiently in the presence of the exponential number of possible error combinations?

Also, it's not clear to my why bounding the delta product from above bounds the errors: very small deltas (e.g. delta=0) applied to 'good' transitions may be as bad - don't they?

line 166 - "states that the worst cannot happen at each time" - it sounds more accurate to say "*can* state that the worst...", since if the budget is large enough, the worst can happen at each time

- While proofs can be found in the supplementary material, it seems necessary to have at least proof sketches inside the paper, especially for proposition 1, and

especially given that some proofs are part of the main claimed contributions of the paper.

- line 177 - if you claim the comparison is 'instructive', it seems appropriate, to guide the reader and provide a brief explanation to what is instructive in the comparison.

- Theorem 2: it might add to clarify if 'Z' is replaced by something line \mathcal{Z}_t

- experiments:

- The experiments verify that your algorithm behaves as expected with

respect to risk and model errors on a grid world example. This is

good as a first step, but it would be better verify your algorithm

on a more complex domain.

- it seems that you built the obstacles in a way that the length of

the optimal path for any given (increasing) safety level is

increasing. It would be good explicitely clarify that (I didn't find

a mention of that in the paper).

- initially it confuses that the obstacles are yellow, given that the

value function is also yellow in parts - it seems like obstacles have

high value

- shouldn't M be negative (since your plotted value function has

negative values)

Summary: The paper proves a novel connection between risk-sensitivity and robustness for a

a CVaR objective. The paper is interesting and well presented. Points that require

improvement are:

(1) providing at least proof sketches inside the paper, especially given

that the proofs are claimed as a main contribution

(2) The paper heavily relies on an error budget constraint that

seems non-intuitive/non-realistic to me. It would be good to

explain whether this error-budget just makes the math easier, or

does it also have a realistic interpretation.

(3) Experiment with a more complex domain could strenghten the paper

Author Feedback
Author rebuttal: We would like to thank the reviewers for thoroughly reading our paper.

All comments and suggestions will be addressed in the final version of this paper. In particular:

To reviewer 1:
Adaptive interpolation: there is a simple trick to perform adaptive interpolation without recalculating V_t. Note that the resolution requirement is y_{i+1}/y_{i} <= \theta. Therefore, it's sufficient to only add points between y'_2 and y_2 (under the requirement y_{i+1}/y_{i} <= \theta, which is trivial to code), and keep the other points unchanged. Since all the additional points now belong to the same original interpolation segment (y_1,y_2), the linear interpolation of V_t does not change, and may be used 'as is' when calculating V_{t+1}. Thank you for pointing this out - we will clarify this issue in the algorithm, and also modify the notation.

To reviewer 2:
The choice of \alpha=0.11 is due to a log-space partitioning of [0,1] to 21 points, and has no special meaning. To obtain the results for \alpha=0.1, one can either interpolate the corresponding value functions of 2 adjacent alphas or include alpha=0.1 in the set of interpolation points.

To reviewer 3:
The discussion on the relation between CVaR and budget-constrained MDP not only motivates our work, but also suggests an important link between the robust MDP framework in [13] and the dual representation of CVaR. A similar study for dynamic risk can be found in the paper "Robustness and risk-sensitivity in Markov decision processes" by Osogami (2012). The static risk counterpart is subtler and models a less conservative situation.

Our methods indeed require explicit knowledge of the MDP while the CVaR policy gradient algorithms in [4,5,8,25] generally do not. We will tone down the comparison. Thank you for pointing out the additional reference; we will duly cite it.

To reviewer 4:
We will add a proof sketch and the relevant discussions to the final version of the main paper.

We comment on the error budget.

Modeling parameter uncertainty (or, equivalently, model errors) in *sequential* decision problems is an open problem that is actively being investigated in reinforcement learning, operations research, and stochastic control.

The most common, but naive approach, is the rectangular uncertainty model. In this model each state is treated independently, and model uncertainty *for each state* is independently measured using statistical confidence intervals on the transition and reward distributions observed from the data. The error in the total reward is then calculated by assuming *the worst-case* parameter realization (within the confidence intervals) at each state. As is well-known, this approach is often too conservative to be of practical use.

A different approach is to assume a statistical distribution on *the errors* (i.e., on the probability of having an error in a state, or on its possible magnitude). This idea was explored in [13], and it was shown, using standard concentration inequalities, that when the error probabilities are IID, then the *number* of errors in a given trajectory is bounded. Similarly, if the error magnitudes are IID, then the sum of error magnitudes - *the error budget* -- is also bounded.

A third approach is to adopt a Bayesian view, and assume a distribution on the *parameter values*, instead of errors induced by an adversary.

The error budget introduced in our paper resembles the approach of [13]. However, further work is required to rigorously relate it to a *statistical model* of parameter uncertainty. We hope that the CVaR connection we discovered will drive such future research; it is an interesting and important topic, which we intend to pursue. Furthermore, we believe that our results may extend beyond CVaR to general coherent risk measures, which may offer additional flexibility in modeling parameter uncertainty.

Regarding the upper bound on the Delta product, note that in Lines 161-162 the perturbations are required to be probability distributions (i.e., positive and sum to 1). Thus, the upper bound is also implicitly a lower bound. In particular, for eta=1, the smallest possible value of eta, there is no perturbation.